# Influence of DLC Coatings Deposited by PECVD Technology on the Wear Resistance of Carbide End Mills and Surface Roughness of AlCuMg2 and 41Cr4 Workpieces

**Sergey N. Grigoriev** , **Marina A. Volosova \*, Sergey V. Fedorov and Mikhail Mosyanov**

Department of High-Efficiency Machining Technologies, Moscow State University of Technology STANKIN, Vadkovskiy per. 3A, 127055 Moscow, Russia; s.grigoriev@stankin.ru (S.N.G.); sv.fedorov@stankin.ru (S.V.F.); mmosyanov@yandex.ru (M.M.)

**\*** Correspondence: m.volosova@stankin.ru; Tel.: +7-916-308-49-00

**Abstract:** The primary purpose of this work was to study the effectiveness of using diamond-like coatings (DLC) to increase the wear resistance of carbide end mills and improve the surface quality of the processed part when milling aluminum alloy and low-carbon steel. The functional role of forming an adhesive sublayer based on (CrAlSi)N immediately before the application of the external DLC film by plasma-enhanced chemical vapor deposition (PECVD) technology in the composition of a multicomponent gas mixture containing tetramethylsilane was established in the article. The article shows the degree of influence of the adhesive sublayer on important physical, mechanical, and structural characteristics of DLCs (hardness, modulus of elasticity, index of plasticity, and others). A quantitative assessment of the effect of single-layer DLCs and double-layer (CrAlSi)N/DLCs on the wear rate of end mills during operation and the surface roughness of machined parts made of aluminum alloy AlCuMg2 and low-carbon steel 41Cr4 was performed.

**Keywords:** diamond-like coatings; nitride sublayer; index of plasticity; adhesive bond strength; end mills; hard alloy; wear resistance; milling of aluminum alloys; milling of structural steels; surface quality

## 1. Introduction

The carbide end mills are the most popular and versatile tools for processing a wide range of metals, alloys, and non-metallic materials [1–5]. The outstanding capabilities of modern carbide mills are provided by a variety of original design solutions, the correct selection of tool geometry, and the use of a wide range of modern wear-resistant coatings [6–8]. The application of wear-resistant coatings with a thickness of 3–7 μm to the working surfaces of end mills gives them the characteristics necessary for specific processing conditions and to ensure that the coating effectively complements the physical and mechanical properties of the hard alloy substrate and together, they will have increased wear resistance during cutting [9–11]. Among the numerous coatings, thin-film diamond-like coatings (DLC) represent a separate group of particular interest. Due to their excellent anti-friction properties and good resistance to abrasive wear, these coatings are now successfully used in mechanical engineering and metalworking as surface protection of machine parts operating under conditions of increased friction with mating parts and cutting tools made of high-speed steels and hard alloys, processing non-ferrous metals and alloys, composite materials, and others [12–16].

For the formation of DLCs of various structures and properties, research organizations and manufacturing enterprises are currently well equipped and have technological equipment based

on the principles of physical vapor deposition (PVD), chemical vapor deposition (CVD), as well as their combination [17–22]. However, whatever method of production we use, the structure of the formed DLC assumes the simultaneous presence of various forms of carbon in them, which can exist in different hybridizations. Modern technologies for the deposition of DLCs provide the formation of films with a different ratio between $sp^3$ (diamond) and $sp^2$ (graphite-like) hybridizations combined in an amorphous structure (Figure 1). The properties of the formed coating depend on the type of bonds that hold the carbon atoms.

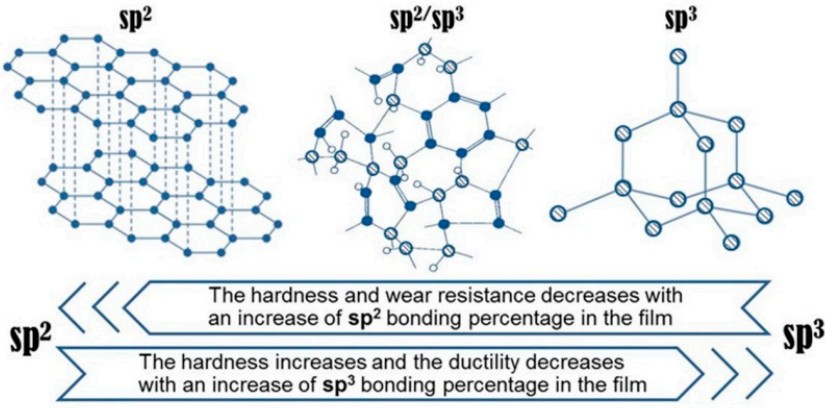

**Figure 1.** Different hybridizations of carbon in diamond-like coatings (DLC) films.

The higher the proportion of $sp^3$ bonds, the closer the properties of such material is to diamond—increase in its hardness and resistance to abrasive wear. In turn, the higher the content of $sp^2$ bonds, the closer the properties of the material are to graphite—its hardness and wear resistance are reduced at temperatures that occur during cutting. When creating DLCs, the main task of technologists is to get a coating with a structure in which the number of $sp^3$ links is increased compared to $sp^2$ links. Furthermore, of great importance for the characteristics of coatings is the amount of hydrogen and other components that are always present in coatings deposited by industrial technologies [23–27].

DLCs have specific features: with an increase in film thickness, the level of residual stresses increases noticeably, the strength of its adhesive bond with the substrate decreases, and, as a result, the coating does not effectively resist the loads acting on it during the cutting of materials (especially cyclic loads during interrupted cutting, for example, milling).

According to the known physical laws, a film of thickness $h$ will peel off when the elastic energy per volume unit due to stress $\sigma$ exceeds the energy required for the formation of two new surfaces by exfoliation [27,28]:

$$h < \frac{4\gamma E}{\sigma^2} \tag{1}$$

where $\gamma$ is surface energy (J/m$^2$), $\sigma$ is stress (Pa), $h$ is the coating thickness (µm), and $E$ is the modulus of elasticity (Pa).

This dependence sets the upper limit of the film thickness, above which it peels off spontaneously. When a tool with a coating is exposed to power loads, the stresses from the external load are additionally superimposed on the residual stresses in the coating, and accordingly, the destruction occurs at significantly lower loads.

Undoubtedly, the stress state of a DLC is influenced by the essential physical and mechanical characteristics—hardness (*H*), modulus of elasticity (*E*), as well as their ratio (*H/E*), called "index of plasticity". There is one more feature that must be taken into account when developing coatings for a hard alloy tool—due to an abrupt change in the modulus of elasticity of the coating layer and the surface layer of the tool material, significant tangential shear stresses arise at their interface, which can

lead to peeling of the coating during cutting. Therefore, to reduce technological stresses at the interface and ensure increased strength of the adhesive bond between the coating and the tool, it is necessary to provide, as close as possible, the characteristics of the modulus of elasticity of these materials [29–34].

To minimize the problems described above, various technological approaches are used when creating DLCs: they are doped with different elements and compounds (silicon, tungsten, and others), and can form adhesive sublayers (intermediate coatings) [35–40]. A universal technical solution cannot be found. When developing the architecture of a DLC and the technology of its deposition, it is necessary to be guided by the specific operating conditions of the coated product, and it is necessary to evaluate the effectiveness of the chosen approach under the conditions of the action of real operating loads or as close as possible to them.

In this work, the goal was to comprehensively study the effect of the formation of an adhesive sublayer based on (CrAlSi)N on carbide end mills before applying the outer DLC film using the PECVD method in the presence of a multicomponent gas mixture containing tetramethylsilane, in comparison with applying only a single-layer DLC, and determining the degree of influence of the adhesive sublayer on the critical physical, mechanical, and structural characteristics of the DLC. Furthermore, we quantified the influence of single-layer (without sublayer) and double-layer (with sublayer) DLCs on the wear rate of end mills during operation and the surface roughness of machined parts made of aluminum alloy AlCuMg2 and low-carbon steel 41Cr4. For evaluating the effectiveness of the DLC under different heat and power loads, we specifically selected two fundamentally different processed materials for research. Compared to other structural materials, aluminum alloys are well amenable to machining, relatively low power and temperature loads on the cutting tool arise. At the same time, they have specific features—repeated adhesion of aluminum particles to the cutting edge of the tool during the cutting process, followed by their tearing out from the working surfaces of the tool, which intensifies tool wear and reduces the quality of the processed surface [41–44]. When machining low-alloy structural steels by end mills, the thermal load is noticeably higher, and the main problem is the wear on the flank of the tool as a result of abrasion caused by hard components in the work material. Furthermore, adhesion wear of the working surfaces of the end mill is observed due to the local grasp of the processed and tool materials, followed by separation of the smallest particles from the tool, which are carried away by the descending chips [45,46].

A separate comment is required to avoid doubt about the validity of the authors' choice of the DLC for the tool processing ferrous steel. According to classical concepts, at elevated temperatures, carbon is intensely dissolved in iron, and graphitization of diamond crystals occurs. However, in this case, the authors do not use a diamond tool but study the behavior of a multicomponent DLC film containing various modifications of carbon, and compounds based on silicon, which is a component of the gas mixture during condensation of the DLC film. We purposefully selected iron-containing steel as the material to be processed to study the functioning of the adhesive sublayer and the outer DLC layer under the loads typical for milling this material. We should add that the authors have research experience in using DLCs for turning hardened bearing steels, and in their previous works, a specific effect was observed from the use of coatings for ceramic tools [21,47–49]. This makes it possible to use a carbide tool as an object of research in this work.

The choice of (CrAlSi)N as an adhesive sublayer material for the functioning of the DLC is not accidental and is explained by the great potential of using this compound for the needs of tool production [49–51]. This nitride coating compares favorably with traditional nitride films such as (TiAl)N, (TiCr)N, and (TiNbAl)N since it is a nanocomposite, its deposition does not entirely mix the components but forms two phases. Its structure consists of AlCrN nanocrystals embedded in an amorphous SiN matrix, which provides a high level of strength of interatomic bonds between atoms of the amorphous and crystalline phases [52,53]. The introduction of silicon into the coating reduces the internal residual stress at the "hard alloy-coating" interface, and the formation of oxides by the coating components at high temperatures slows down the tribochemical reactions on the contact surfaces of the cutting tool. These features, and previously obtained experimental data [49,54,55], allow us to

count on the effectiveness of the composite compound (CrAlSi)N for carbide end mills as an adhesive sublayer before applying the external DLC.

## 2. Materials and Methods

### 2.1. Processed Materials, Cutting Tools, and Operational Testing Methods

As the processed material, two types of materials widely used in mechanical engineering were chosen—aluminum alloy AlCuMg2 and structural low-alloy steel 41Cr4, the chemical composition of which is given in Tables 1 and 2.

**Table 1.** Chemical composition of the AlCuMg2 alloy used for testing.

| Element | Al | Cu | Mg | Mn | Fe | Si | Zn | Ti | Cr | Impurities |
|---|---|---|---|---|---|---|---|---|---|---|
| Content (%) | 92.0 | 4.3 | 1.6 | 0.7 | 0.4 | 0.5 | 0.25 | 0.15 | 0.1 | 0.15 |

**Table 2.** Chemical composition of the 41Cr4 steel used for testing.

| Element | Fe | Cr | Mn | C | Si | S | Cu | P |
|---|---|---|---|---|---|---|---|---|
| Content (%) | 97.0 | 1.2 | 0.9 | 0.44 | 0.37 | 0.034 | 0.032 | 0.024 |

The processed materials used in work differ significantly in their physical and mechanical characteristics—the aluminum alloy AlCuMg2 has a tensile strength of 245 MPa with a hardness of 105 HB, and 41Cr4 steel, 635 MPa and 210 HB, respectively. Such differences predetermine a significant difference in the heat and power loads that act on the cutter's cutting edge during cutting. When machining 41Cr4 steel, more than a twofold increase in the component of the cutting force Pz and the temperature in the cutting zone is observed.

As cutting tools for carrying out a set of experimental studies we used 3-flute (for processing AlCuMg2 alloy) and 5-flute (for processing 41Cr4 steel) end mills with a diameter of 6 mm, made of tungsten-cobalt hard alloy 6WH10F (WC, 90%; Co, 10%) with hardness HRA 92.1 and density 14.50 g/cm$^3$. Figure 2 shows a general view of the design of the end mills, and Table 3 shows their design and geometric parameters.

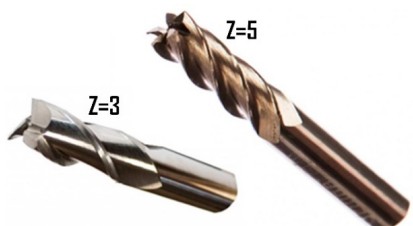

**Figure 2.** Design of end mills used in research.

**Table 3.** Design and geometric parameters of end mills.

| Tool Parameters | Processed Material | |
|---|---|---|
| | **AlCuMg2** | **41Cr4** |
| Mill type | end | |
| Shank type | cylindrical | |
| Material | 6WH10F | |
| External diameter *D*, mm | 6 | |
| Cutter length *L*, mm | 36 | 40 |
| Working unit length *l*, mm | 16 | |
| Number of teeth *Z* | 3 | 5 |
| Shank diameter *d*, mm | 6 | |
| Inclination angle of the chip grooves ω | 30° | |

To perform a complex of metallographic and metallophysical studies, square plates with dimensions of 12.0 mm × 12.0 mm × 5.0 mm were manufactured of 6WH10F hard alloy.

Operational tests of the end mills were performed at the CTX beta 1250 TC turning/milling centre (DMG MORI Co., Ltd., Bielefeld, Germany). An ER collet chuck made according to DIN 69893-1 HSK-A63 was used to clamp the end mills. The workpieces were milled in rods with a diameter of 65 mm of aluminum alloy AlCuMg2 and structural low-alloy steel 41Cr4. In the experiments, the coated end mills were tested without coolant to also provide better observation of wear, and that correlates with standard tool tests. Tool tests were carried out using a program written in the SINUMERIK 840D SL system (Siemens AG, Regensburg, Germany) while executing the strategy of milling the plane of the workpiece end face in the following cutting modes (Figure 3):

- for AlCuMg2 alloy—cutting speed $Vc$ = 188.4 m/min, feed rate $F$ = 2000 mm/min, feed per tooth $Fz$ = 0.068 mm/tooth, milling width $B$ = 2 mm, and milling depth $t$ = 0.5 mm;
- for 41Cr4 steel—cutting speed $Vc$ = 188.4 m/min, feed rate $F$ = 750 mm/min, feed per tooth $Fz$ = 0.015 mm/tooth, milling width $B$ = 2 mm, and milling depth $t$ = 0.2 mm.

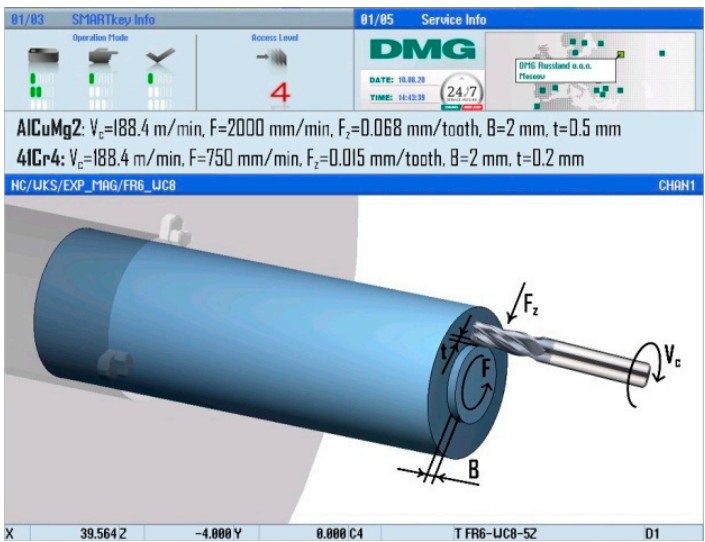

**Figure 3.** Processing strategy and cutting conditions for machining AlCuMg2 and 41Cr4 steel workpieces with end mills.

The limit size of the wear chamfer on the flank surface (0.4 mm) was taken as the criterion for loss of performance (failure) of the end mills. Tool wear resistance was defined as the cutting time until the cutter reached the limit of wear. To quantify wear, we used a metallographic optical microscope Stereo Discovery V12 (Carl Zeiss Microscopy GmbH, Jena, Germany); the milling cutters were placed in a particular device at an angle of 45° on its table. Each tooth of the cutter was measured, and the largest amount of wear was detected, and this was taken into account when processing the results of experiments and constructing curves for the dependence of wear on the flank surface of the cutting time, based on which the resistance value was calculated.

When processing 41Cr4, one pass of the cutter was 3.5 min, and after each pass, the wear chamfer was measured. When processing AlCuMg2, one pass of the cutter was 3 min, and taking into account that the flank surface wear is less intense when processing aluminum alloys, wear was measured every three passes. After each cycle of experiments, a disk was cut off from the workpiece, and its end surface, previously treated with an end mill, was examined on a Hommel Tester T8000 stylus profiler by JENOPTIK Industrial Metrology (VS-Schwenningen, Germany) to quantify the surface roughness.

### 2.2. Technology and Equipment for the Deposition of DLC Coatings on End Mills

To apply DLCs to the surface of the end mills, they were placed on the turntable of the vacuum chamber of a hybrid technological unit (Figure 4), the design of which allows the sequential application of coatings in a single cycle by various methods: vacuum-arc deposition by evaporation of cathode materials (Arc-PVD); plasma-chemical gas-phase deposition in a glow discharge plasma by chemical reaction and decomposition of gas mixture components-plasma-enhanced chemical vapor deposition (PECVD). Two types of coatings were applied: (1) double-layer coatings consisting of an adhesive sublayer (CrAlSi)N and the outer DLC layer; (2) single-layer DLCs.

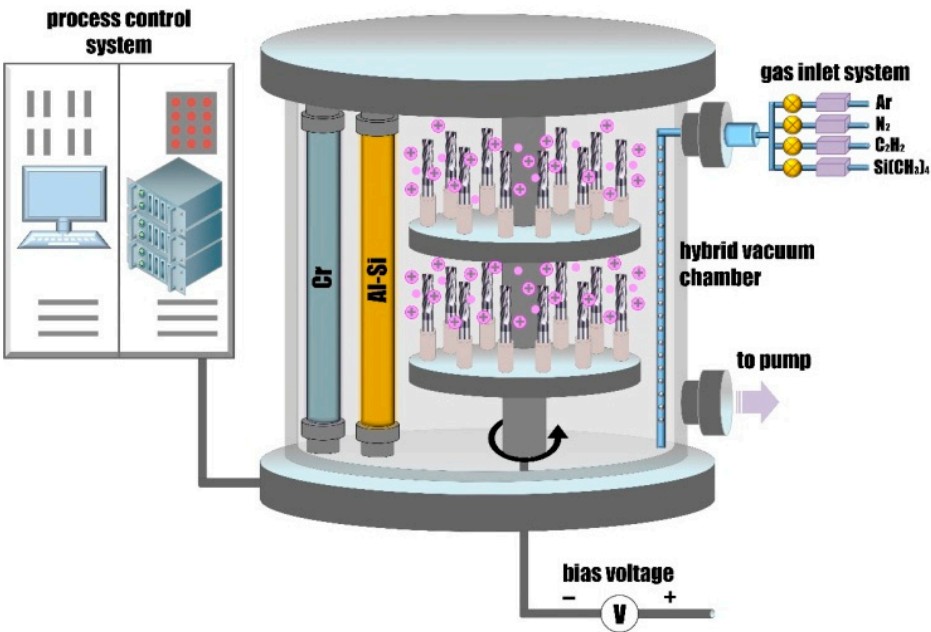

**Figure 4.** The principal application of DLC-coatings and (CrAlSi)N/DLC-coatings on end mills with the use of hybrid technology unit.

We used the original PI311 series technological unit developed by PLATIT AG (Selzach, Switzerland). The unit (Figure 4) is equipped with a gas filtration device and a multi-channel gas inlet system, a vacuum pumping system, a source of bias voltage to the turntable, and two cylindrical rotating cathodes made of chromium (Cr) and silumin (AlSi). For minimizing micro drops in the formed coating (CrAlSi)N, the unit implements the original lateral rotating cathodes (LARC) technical solution. This technology does not involve the use of bulky magnetic separators to minimize micro drops but uses rotating cathodes that are placed on the periphery of the chamber, and virtual gates that work without any mechanical elements. Following the generally accepted classification and terminology, the DLCs deposited in this work refer to hydrogenated diamond-like a-C: H films.

Preliminary cleaning of the cutters, which is necessary to achieve adequate strength of the adhesive bond of the deposited coating, was carried out with argon ions with an energy of 500 eV at a pressure of 1 Pa using a non-self-sustaining gas discharge ignited between the cathodes. At the same time, a negative voltage of 400 V was applied to the rotary table with the processed samples. The electron flow between the cylindrical cathodes created a high-density plasma, by which the samples were efficiently cleaned of impurities and oxides for 15 min before the deposition of the coating. After the cleaning was completed at a pressure of 1.5 Pa, gradually decreasing to 0.7 Pa, an adhesion sublayer (CrAlSi)N was formed by two Cr and AlSi cathodes when a gas mixture of nitrogen (volume fraction 90%) and argon (volume fraction 10%) and supplying negative voltage to the turntable of 500 V. The time of deposition of the (CrAlSi)N layer was 25 min, which provided a thickness of about 1.5 μm. Then the outer diamond-like layer was formed by plasma-chemical gas-phase deposition in a glow

discharge plasma by starting a chemical reaction and decomposing the components of the gas mixture supplied to the chamber: acetylene $C_2H_2$ (volume fraction 78%), argon Ar (volume fraction 7%), and tetramethylsilane $Si(CH_3)_4$ (volume fraction 5%). The deposition time of the diamond-like layer was 180 min, which provided a thickness of about 2.5 µm. For the application of a single-layer DLC, the process step associated with the formation of an adhesive sublayer was excluded, and after cleaning the cutters, the DLC layer was deposited directly. The DLC layer was deposited at a pressure of 4.0 Pa. This value's choice is because a decrease in this parameter leads to a decrease in the productivity of the process when an increase leads to an excessive increase in the structure of the graphite-like component's coating.

A Tescan VEGA3 LMH scanning electron microscope was used to analyze the structure of DLCs deposited using the technology described above.

### 2.3. X-ray Photoelectron and Diffraction Analysis of DLCs

X-ray photoelectron spectroscopy (XPS) was used to study the chemical and electronic state of carbon atoms on the surface of samples with DLCs. The analysis was performed using Thermo Scientific's K-ALPHA X-ray photoelectron spectrometer (Thermo Fisher Scientific Inc., Bremen, Germany).

The samples were both shot in the initial state of the coating surface and after surface treatment for 6 min with an $Ar^+$ ion beam to remove the adsorbed layers with high carbon and oxygen content. High-resolution photography was performed using monochromatic and polychromatic Mg K$\alpha$ and Al K$\alpha$ radiation at a power of 200–300 W and a voltage of 14 kV, pressure in the chamber $7 \times 10^{-8}$ Pa.

XPS analysis was performed for various peaks of the DLC layer within the prominent C1s peak: $sp^2$ and $sp^3$ peaks, characterizing carbon hybridization, and C=O and C–O peaks, characterizing various compounds of the components with oxygen and other elements. Besides, we analyzed the peaks of the phases containing silicon compounds in the coating. For each component of the DLC layer, the following characteristics were measured using XPS: start binding energy (Start BE), binding energy (BE), end binding energy (End BE), full width at half maximum (FWHM), and atomic percent (Atomic %).

A Panalytical Empyrean X-ray diffractometer (PANalytical, Almelo, The Netherlands) with monochromatic Cu K$\alpha$-radiation was used to study the stresses. The stresses were determined by the classical $\sin^2\psi$ method with a grazing beam at a fixed angle of incidence of the beam $\psi_0$ and scanning along the 2$\theta$ axis.

### 2.4. Microhardness, Modulus of Elasticity, and Adhesive Bond Strength of DLCs

We used the method of nanoindentation with a Berkovich diamond indenter on CSEM Nano hardness tester (CSM Instruments, Needham, MA, USA) equipped with specialized software (version 4.0) based on the algorithm of Oliver and Pharr [26,56]. This method, in contrast to the Vickers pyramid microhardness estimation scheme, is more reliable, since, if necessary, it allows excluding the influence of the hardness of the underlying layers on the results of measurements of the characteristics of the outer DLC layer. Before starting the measurements, the equipment was calibrated on reference samples with the known modulus of elasticity and hardness.

In experiments, the value of the applied load and the corresponding depth of indenter insertion were selected based on the condition of not more than 15% of the thickness of the DLC. The measurements were performed at loads of 1.0 and 4.0 mN, and the corresponding maximum indenter insertion depths were 0.09 and 0.38 µm, respectively. The tests were carried out as follows: after the load on the coated sample reached the maximum value, unloading began, and the load acting on the indenter gradually decreased to zero. The duration of the load-unload cycle was 50 s. According to the measurement results, typical experimental curves of continuous indentation were obtained—"the dependence of the load on the depth of indentation". The first curve corresponds to loading and reflects the resistance of the material to the penetration of the indenter. The second describes the return of deformation after

the removal of the external load and characterizes the elastic properties of the material. In this case, the deformation response (indentation depth) is simultaneously recorded in nanometer resolution.

The obtained experimental data allowed us to not only judge the nano hardness (*H*) but also the modulus of elasticity (*E*) of the DLC. If we obtain a quantitative value of the *H/E* ratio (index of plasticity) [57–59], we can approximate the viscosity of the coating and its ability to resist possible deformation and destruction during cutting. For reducing possible measurement errors, each sample was processed five times, resulting in an average hardness value.

To quantify the strength of the adhesive bond of DLCs to carbide substrates, we used the CSEM micro scratch tester system. It implements the method of scratching with a diamond cone indenter (apex angle of 120° and apex radius of 100 μm) with a variable load from 1 to 30 N. The spectra of the acoustic emission signal were constantly recorded. Their bursts make it possible to reliably judge the initial stage of cracking and subsequent peeling of the coatings. Hard-alloy samples with coatings moved at a constant speed, and the recording and processing of acoustic emission spectra were carried out using specialized software (version 4.0). Thus, when conducting scratch tests, is solved the main problem of determining the critical load, at which there is an abnormal change in the indentation depth of the indenter and the separation of the coating.

In addition to the method described above, a qualitative assessment of the adhesive bond strength of the coating to the substrate was performed by pressing the Rockwell indenter at a load of 15 N/mm$^2$ on a Wilson Hardness R574T device by INSTRON (Norwood, MA, USA). In this case, a comparative assessment of the adhesion characteristics of two types of coatings was made by comparing the prints obtained after the intrusion of a cone indenter.

### 2.5. Friction Coefficient and Abrasion Resistance of DLC Coatings

The friction coefficient was evaluated in work on a Tetra Basalt N2 testing machine from TETRA GmbH to assess the DLC's effect on the transformation of frictional properties. During the tests, the coefficient of friction-sliding of rubbing pairs "hard alloy with DLC-coating—a counter body made of AlCuMg2 alloy" and "hard alloy with DLC-coating—a counter body made of 41Cr4 steel" were determined. The tests of all samples were carried out under conditions of dry friction at identical normal loads on the counter body (1 N), with the speed of relative displacement being 2 mm·s$^{-1}$ and the friction path being 90 mm. We evaluated the coefficient of friction for samples with single-layer DLCs.

The abrasion resistance of the surface layer of DLC-coated carbide samples was carried out by testing on a Calotest unit by CSM Instruments. The test principle was that a rotating sphere was placed on a test sample and operated with a preselected load of 20 N, which was controlled by a particular sensor. The rotation of the sphere during the tests was carried out by the driveshaft, and the position of the sphere to the test samples and the load in the contact area was constant. A water-based abrasive suspension was fed to the test area, and its solid particles in the contact area of the sphere with the coated sample led to abrasion of the surface area and the formation of a wear hole (spherical recess). The suspension and sphere wear the coating and substrate in a controlled manner, which guarantees reproducible results. Optical analysis of the geometric dimensions of the worn hole makes it possible to judge the ability of hard-alloy samples with DLCs to resist abrasion [60,61].

## 3. Results

### 3.1. Structure and Chemical Composition of DLCs

Figure 5 shows SEM images of cross-sections of carbide samples with one-layer and double-layer DLCs. The presented photographs give a general idea of the structure of the investigated coatings and the thickness of their layers formed under the selected combination of technological modes. The thickness of the formed adhesive sublayer (CrAlSi)N was 1.4 ± 0.15 μm, and the thickness of the outer DLC layer was 2.6 ± 0.2 μm.

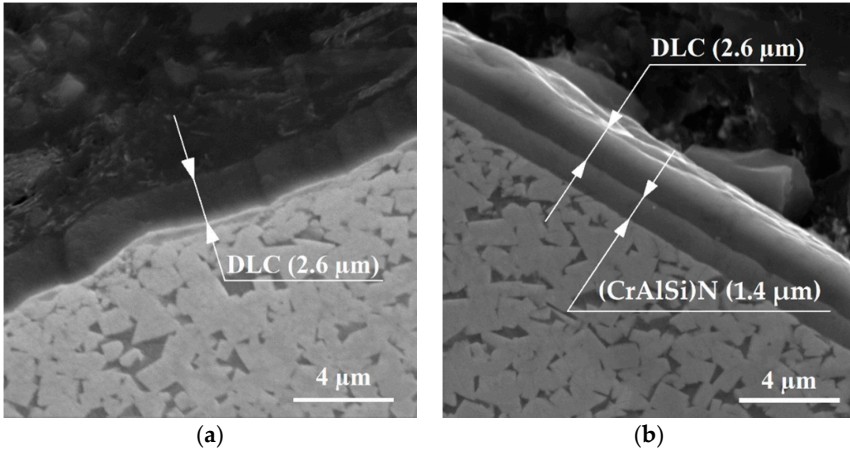

**Figure 5.** SEM images in secondary electrons of cross-sections of carbide samples, high voltage of 10.0 kV, view field of 16.6 μm, 25.0 k×: (**a**) single-layer DLC; (**b**) double-layer DLC.

Tables 4 and 5 present data for processing and decoding the results of XPS analysis of two types of samples with DLCs without sublayer (Table 4) and with sublayer (CrAlSi)N (Table 5). These results were obtained using specialized software of K-alpha X-ray photoelectron spectrometer, an Avantage Data System (version 5.0) (Thermo Scientific, Bremen, Germany). The analysis was performed after the surface layer of the coating with a thickness of about 0.004 μm was etched with argon ions.

**Table 4.** Results of XPS-analysis software interpretation: composition and energy characteristics of DLC peaks (without sublayer).

| XPS-Peak | Start BE | Peak BE | End BE | FWHM eV | Atomic % |
|---|---|---|---|---|---|
| C1$s$ $sp^2$ (graphite) | 297.98 | 284.5 | 279.18 | 1.1 | 16.03 |
| C1$s$ $sp^3$ (diamond) | 297.98 | 285.3 | 279.18 | 1.3 | 47.92 |
| C1$s$ C=O | 297.98 | 286.5 | 279.18 | 2.4 | 13.97 |
| C1$s$ C–O | 297.98 | 286.1 | 279.18 | 2.5 | 10.58 |
| O1$s$ C–O | 545.08 | 533.5 | 525.08 | 1.9 | 4.96 |
| O1$s$ C=O | 545.08 | 531.2 | 525.08 | 1.6 | 4.03 |
| Si 2$p^3$ Si, Si–N, Si–C | 110.08 | 101 | 95.08 | 2.7 | 2.51 |

**Table 5.** Results of f-analysis software interpretation: composition and energy characteristics of DLC peaks (with (CrAlSi)N sublayer).

| XPS-Peak | Start BE | Peak BE | End BE | FWHM eV | Atomic % |
|---|---|---|---|---|---|
| C1$s$ $sp^2$ (graphite) | 297.98 | 284.3 | 279.18 | 1.0 | 15.24 |
| C1$s$ $sp^3$ (diamond) | 297.98 | 284.9 | 279.18 | 1.2 | 51.22 |
| C1$s$ C=O | 297.98 | 286.5 | 279.18 | 2.4 | 10.35 |
| C1$s$ C–O | 297.98 | 286.1 | 279.18 | 2.5 | 7.58 |
| O1$s$ C–O | 545.08 | 533.4 | 525.08 | 1.9 | 4.46 |
| O1$s$ C=O | 545.08 | 531.8 | 525.08 | 1.6 | 4.18 |
| O1$s$ SiO$_2$ | 545.08 | 532.5 | 525.08 | 1.5 | 2.95 |
| Si 2$p^3$ Si, Si–N, Si–C | 110.08 | 100.4 | 95.08 | 2.8 | 4.02 |

It should be noted that the binding energies of the coating components are very close for both coatings. The electron energy spectra of the primary coating peak (C1$s$) indicate that in the studied DLCs chemical bonds prevail between carbon atoms with a binding energy of about 284–285 eV, representing $sp^2$- and $sp^3$-hybridized states. The percentage of the diamond component ($sp^3$ hybridization) in coatings for a sample with a DLC without a sublayer is 47%, and 51% for a sample with a sublayer (we emphasize that here we are talking about a pure diamond component). The proportions of graphite-like carbon hybridization ($sp^2$) present in the coatings are very similar for the two samples

studied, 16% and 15%, respectively. Besides, in the central peak of the C1*s* coating, the analysis revealed the presence of various impurities and contaminants, which is typical for DLC films. The percentage of this component for a sample with a DLC without a sublayer is 24%, and 17% for a sample with a sublayer. According to authoritative scientists' works, when choosing rational modes of deposition of the investigated type of DLCs, the formation of a diamond phase ($sp^3$ hybridization) in the range of 30–50% is characteristic [10,20,24,26]. That is, the results obtained are at the upper limit of the possible range of values.

XPS analysis of DLC films revealed various oxygen forms with increased binding energy values (O1*s* peak), 531.8 eV and 533.5 eV. Both samples contain oxygen in the form of surface adsorbed groups, and the proportion of the oxygen component is approximately the same and is about 9%. The only difference is that in a DLC film deposited on a sublayer (CrAlSi)N, in the primary oxygen peak of O1*s*, another component was found, identified as silicon dioxide $SiO_2$, its proportion is slightly less than 3%. A photoelectron peak of silicon Si $2p^3$ was also detected when examining the surface of the samples. The proportion of identified silicon compounds (such as Si, Si–N, SiC) for the studied samples was almost identical and amounted to 2.5–3.0%.

### 3.2. Microhardness, Modulus of Elasticity, Residual Stress and Adhesion Strength of DLCs

Table 6 shows the results of hardness measurements according to the Martens scale (HM), which characterizes both elastic and plastic properties of DLC-coated carbide samples. As follows from the experimental data obtained, with an increase in the applied load and, accordingly, with an increase in the penetration depth of the indenter, the hardness of the surface layer decreases. Within one load (1.0 mN), the hardness of the coated samples varies from 32 to 38 GPa, and from 23 to 29 GPa at a load of 4.0 mN, respectively. Simultaneously, the microhardness value is practically identical for samples with DLCs without a sublayer and with a (CrAlSi)N sublayer. Another type of change is observed when evaluating the modulus of elasticity of samples with DLCs. If at a load of 1.0 mN, the modulus of elasticity of the two types of samples had similar values, 284–300 GPa for samples with DLC without sublayer and 275–295 GPa for samples with (CrAlSi)N/DLC-coated, then with an increase in the load to 4.0 mN, significant differences were observed of 235–239 and 193–209 GPa, respectively. The data in Table 6 also gives an idea of the index (coefficient) of the plasticity of the studied samples. At an applied load of 1.0 mN, the index of the plasticity of the samples are very close and is 0.113 for samples without sublayer and 0.119 for samples with (CrAlSi)N/DLC, and with a four-fold increase in the load to 4.0 mN, the differences are quite noticeable, 0.10 and 0.129 respectively.

**Table 6.** Physical and mechanical characteristics of DLCs deposited on hard alloy samples.

| No. | Type of Test Sample | Applied Load (mN) | Penetration Depth, (nm) | Hardness HM (GPa) | Modulus of Elasticity *E* (GPa) | Index of Plasticity (*H/E*) |
|---|---|---|---|---|---|---|
| 1 | Hard alloy/DLC | 1.0 | 90 | 33 ± 3 | 292 ± 8 | 0.113 |
| 2 | Hard alloy/(CrAlSiN/DLC | | | 34 ± 2 | 285 ± 10 | 0.119 |
| 3 | Hard alloy/DLC | 4.0 | 380 | 25 ± 2 | 242 ± 7 | 0.10 |
| 4 | Hard alloy/(CrAlSiN/DLC | | | 26 ± 3 | 201 ± 8 | 0.129 |

It should be noted that the obtained value correlates to the known data of other research groups and show that the researched DLC is not fragile since the measured range of values for hardness is relatively low when sublayer increases its plasticity.

As follows from the results of the analysis of samples on an X-ray diffractometer (Figure 6), the character of the distribution of residual compressive stresses in the coating for samples with a

single-layer DLC and a (CrAlSi)N/DLC do not change significantly. Attention is drawn to a noticeable decrease in residual stresses' average values when a DLC is applied to a sample with a (CrAlSi)N sublayer. Average residual stresses for single-layer DLCs are 1250–3650 MPa, while 650–2750 MPa for DLCs with a sublayer.

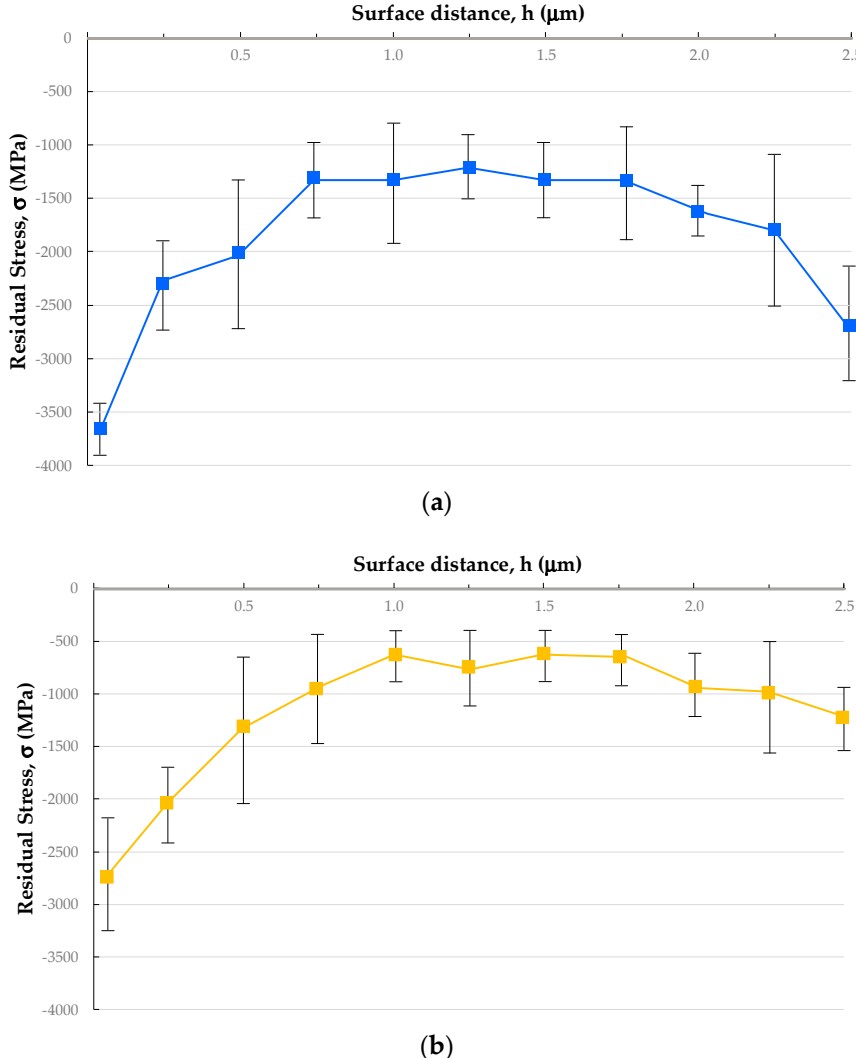

**Figure 6.** Values of residual stresses in DLCs deposited on different samples: (**a**) coating without sublayer and (**b**) coating with sublayer (CrAlSi)N.

Figure 7 shows optical images of indenter movement tracks on the surface of hard alloy samples with DLCs and their corresponding loads from 1 to 30 N obtained from scratch testing. The presented panoramas make it possible to quantify the value of critical loads at which the coatings break off from the carbide substrate and on which part of the indenter movement path this occurred. Acoustic emission sensors registered peak loads indicating the beginning of the indenter penetration into the coating ($P1$), the beginning of the first crack ($P2$), the peeling of local sections of the coating from the substrate ($P3$), and the critical (destructive) load ($P4$). It can be seen that, for samples with (CrAlSi)N/DLC, the peak loads are shifted to the right. In particular, the critical load $P4$ corresponds to significantly higher forces (about 23 N), while for a sample with a DLC without a sublayer the critical load corresponds to the force equal to 12 N. It should be noted that to exclude erroneous experimental results and subsequent erroneous conclusions, five samples of each type of DLCs were subjected to scratch testing. The data obtained on the quantitative values of critical loads for the samples under study were similar, and the spread of values was ±2 N.

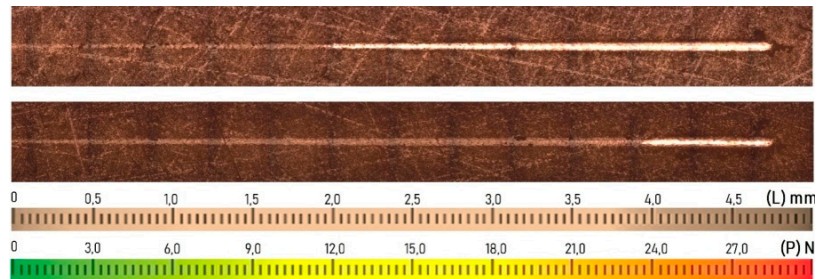

**Figure 7.** Tracks of indenter movements on the surface of 6WH10F hard alloy samples with DLCs and corresponding loads from 1 to 30 N; coating without sublayer (upper image), coating with sublayer (CrAlSi) N (lower image).

Observing such significant differences in the results of scratch testing of samples with DLCs, we additionally performed a qualitative rapid assessment of the strength of the adhesive bond by indentation of the Rockwell indenter. Figure 8 shows the micro images of the indenter insertion area in the samples. It is seen that the penetration of an indenter into a sample with a DLC without a sublayer is accompanied by the exfoliation of micro-sections of the coating in a circle at a certain distance from the indentation area (Figure 8a). In this case, the load applied to samples with (CrAlSi)N/DLC (Figure 8b) does not cause noticeable peeling and destruction of the outer coating.

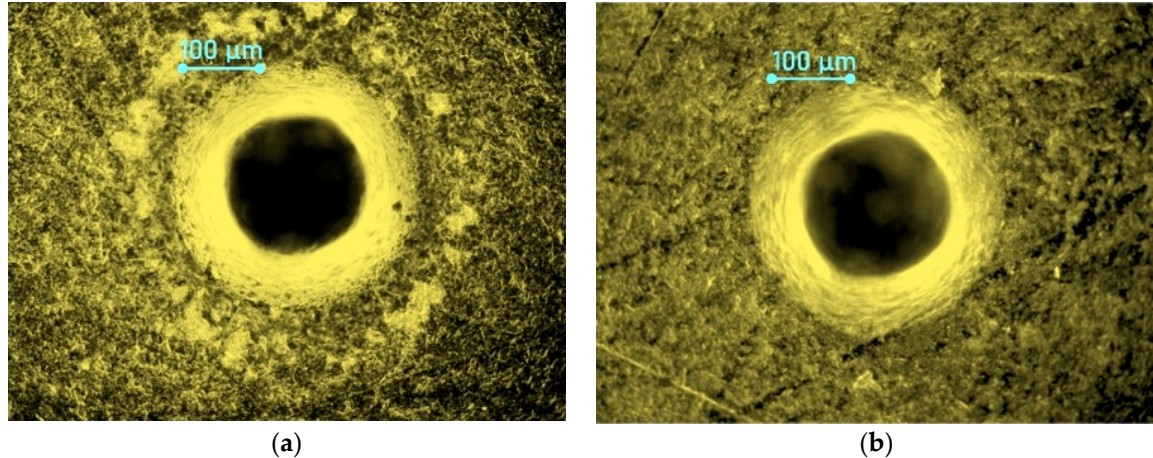

(**a**)                                    (**b**)

**Figure 8.** Rockwell indenter indentation area in the surface layer of 6WH10F hard alloy samples with DLCs: (**a**) coating without sublayer and (**b**) coating with sublayer (CrAlSi)N.

### 3.3. Friction Coefficient and Abrasion Resistance of DLCs

As shown from Table 7, DLC significantly reduces the hard alloy's friction on the aluminum alloy from 0.29–0.32 to 0.15–0.16. The effect of coating deposition on friction against steel is also noticeable, the coefficient of friction coefficient was reduced from 0.41–0.44 to 0.25–0.27.

**Table 7.** Friction coefficient of DLCs deposited on hard alloy samples.

| No. | Type of Sample | Type of Counter Body | Friction Coefficient Value |
|-----|----------------|----------------------|----------------------------|
| 1 | Hard alloy 6WH10F | AlCuMg2 | 0.29–0.32 |
| 2 | Hard alloy 6WH10F/DLC | AlCuMg2 | 0.15–0.16 |
| 3 | Hard alloy 6WH10F | 41Cr4 | 0.41–0.44 |
| 4 | Hard alloy 6WH10F/DLCs | 41Cr4 | 0.25–0.27 |

Figure 8 shows two-dimensional optical images of wear holes on the surface of hard alloy samples after a rotating sphere is forcibly applied to them for 5 min in the presence of an abrasive suspension

in the contact area. When measuring the linear dimensions (*X* and *Y*) of the section of material that has undergone abrasion, we judged the quantitative value of wear and subsequently concluded that the carbide samples are resistant to abrasion. The results shown in Figure 8 show marked differences for three types of samples: uncoated (Figure 9a), DLC-coated (Figure 9b), and (CrAlSi)N/DLC-coated (Figure 9c). There are differences in the shape of the formed wear hole and its linear dimensions. For an uncoated hard alloy sample, the worn surface area has an irregular shape with dimensions (*X* and *Y*) of 0.9 mm × 0.6 mm. For samples with DLC and (CrAlSi)N/DLCs, the worn areas have the shape of spherical recesses with dimensions of 0.6 mm × 0.55 mm and 0.4 mm × 0.4 mm, respectively.

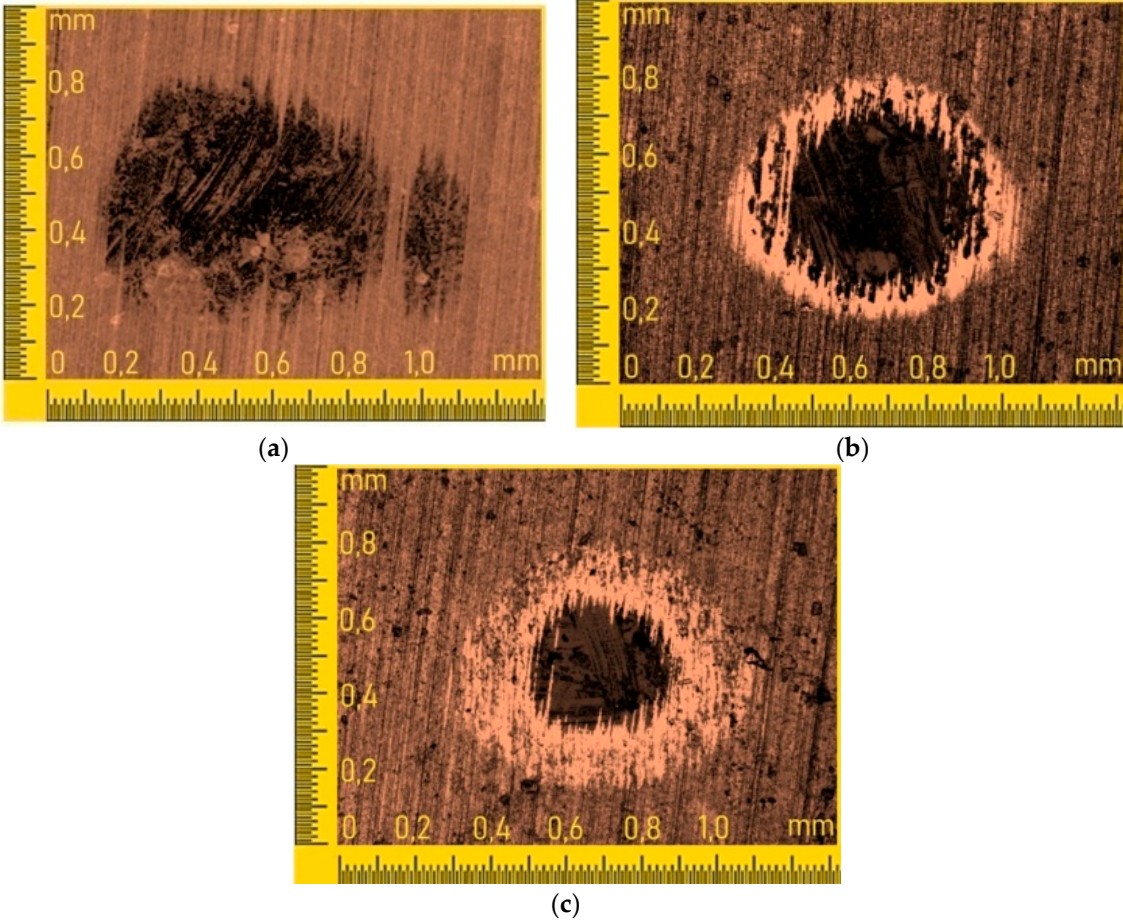

**Figure 9.** 2D images of the worn hole on 6WH10F carbide samples after surface layer abrasion resistance tests: (**a**) uncoated; (**b**) DLC-coated; and (**c**) (CrAlSi)N/DLC-coated.

### 3.4. DLC-Coated End Mill Performance Test

#### 3.4.1. Processing of Aluminum Alloy AlCuMg2

The objects of study during the tests were the wear of three-flute end mills made of 6WH10F hard alloy (Figure 2) and the surface roughness of machined workpieces in the form of bars made of aluminum alloy AlCuMg2.

Figure 10 shows the experimentally obtained dependence of the flank wear chamfer (*h*) of end mills on the tool time (*T*) with various coatings, uncoated, with DLC and (CrAlSi)N/DLC.

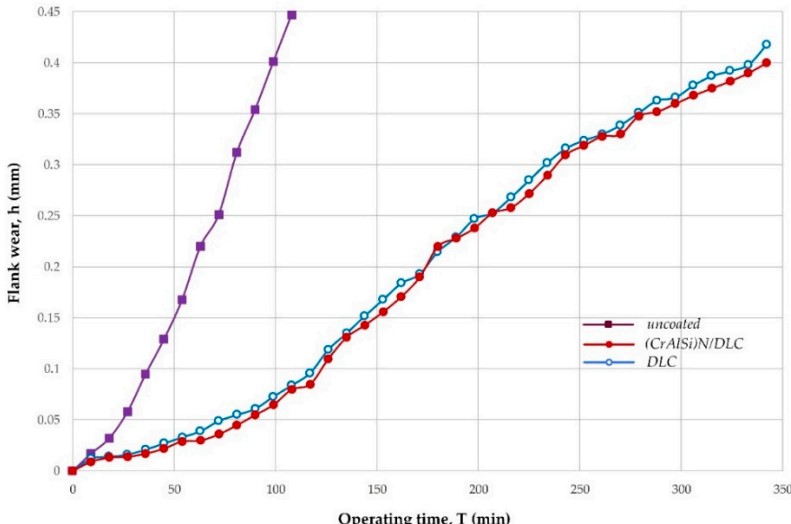

**Figure 10.** Dependence of the wear chamfer size on the flank surface of the end mills on the operating time when machining the aluminum alloy AlCuMg2 with end mills with different coatings.

These data demonstrate a pronounced positive effect of DLCs on the resistance of milling cutters (up to the maximum wear value of 0.4 mm) when processing the aluminum alloy. The development of the wear chamfer on the flank surface due to the coating slows down many times; if the tool life without coating was 99 min, the tool life with DLC is now 342 min. Optical images (Figure 11) showing the general appearance of the worn chamfer on the flank surface of the milling cutters after 72 min of operation, illustrating the above well—the coating significantly slows down the development of wear processes. Thus, we can conclude that the durability of a DLC-coated cutter is 3.45 times higher than that of an uncoated cutter when processing an aluminum alloy AlCuMg2.

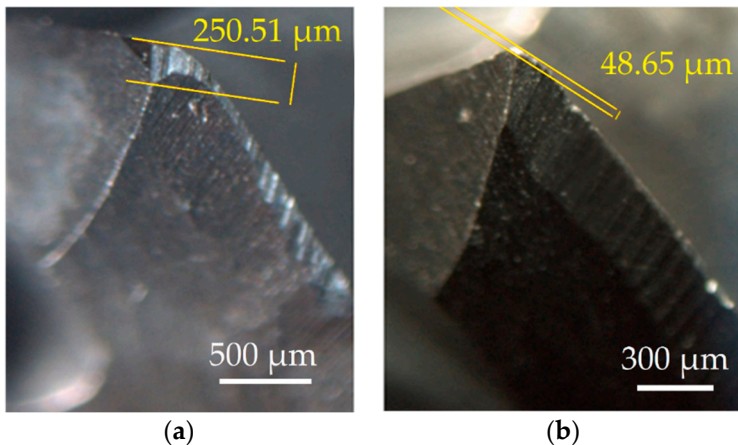

(**a**)             (**b**)

**Figure 11.** The size of flank wear chamfer after 72 min of operation when processing aluminum alloy AlCuMg2 with end mills: (**a**) uncoated, 250.51 μm; and (**b**) with a single-layer DLC, 48.65 μm.

An interesting fact is noteworthy, the application of a (CrAlSi)N sublayer before the formation of the DLC does not have any effect on the cutter's behavior during operation (Figure 10).

The experimental data obtained during the tests (Figure 12) show that when machining an aluminum alloy with uncoated end mills, the surface roughness (*R*) increases significantly after 27 min of operation and at the time of tool failure (99 min), is about 3 μm. When cutting an aluminum alloy with a diamond-like coated cutter, the roughness of the surface of the processed workpiece during 150 min of tool operation practically does not change and, on average, is 1.3 μm. At the moment of failure of the coated tool (342 min), the average roughness value was 2.5 μm. It is important to note

that even in the first minutes of operation of the still unworn cutter, there is a noticeable difference in the surface quality of the workpiece, achieved by the tool without coating and with DLC of 1.75 µm and 1.25 µm, respectively. Experimental data presented in Figure 12 show that the deposition of a sublayer (CrAlSi)N before forming the DLC does not significantly affect the change in the roughness of the treated surface during the end mill operation.

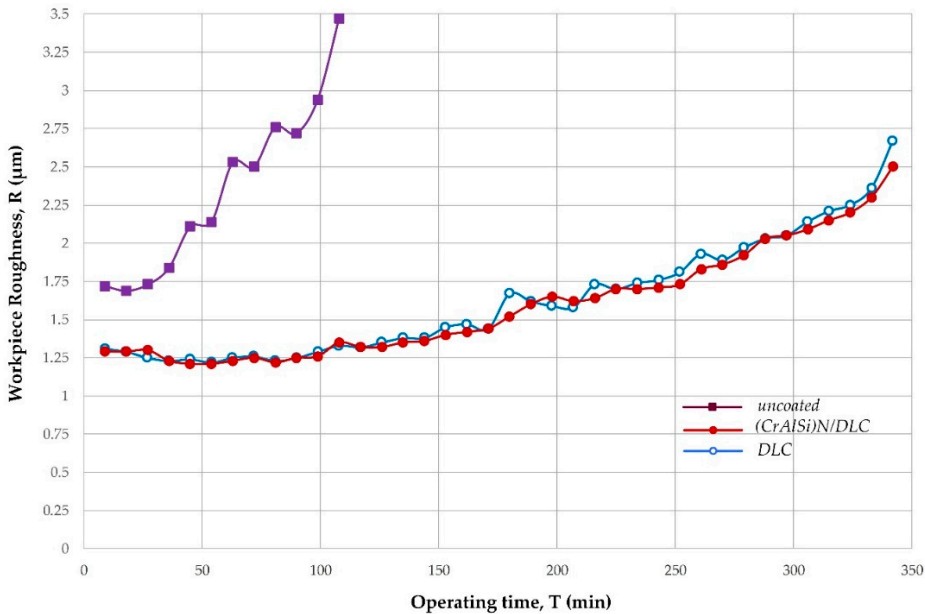

**Figure 12.** Dependences of the roughness of the treated surface of a workpiece made of aluminum alloy AlCuMg2 on the operating time of end mills with different coatings.

### 3.4.2. Processing of Structural Steel 41Cr4

The objects of study during the tests were the wear of five-flute end mills made of 6WH10F hard alloy (Figure 2) and the surface roughness of workpieces in the form of bars made of structural steel 41Cr4. As can be seen from the experimental data presented in Figure 13, in the process of machining structural steel, rather intense wear of the uncoated end mills occurs; the operating time to failure (until the wear limit value of 0.4 mm is reached) is slightly more than 30 min. Simultaneously, the application of a single-layer DLC to the cutter has little effect on the wear rate of the tool and only slightly increases its tool life (no more than 4 min). Optical images (Figure 14), showing the general view of the worn chamfer on the flank surface of the cutters after 17.5 min of operation, well illustrate the above well—the coating application practically does not slow down the development of wear processes.

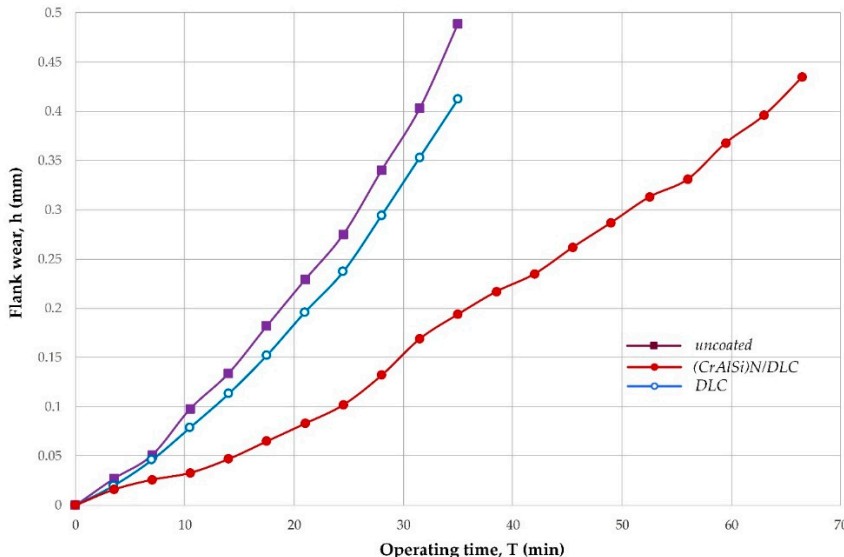

**Figure 13.** Dependences of the wear chamfer size on the flank surface of the end mills on the operating time when machining 41Cr4 structural steel with end mills with different coatings.

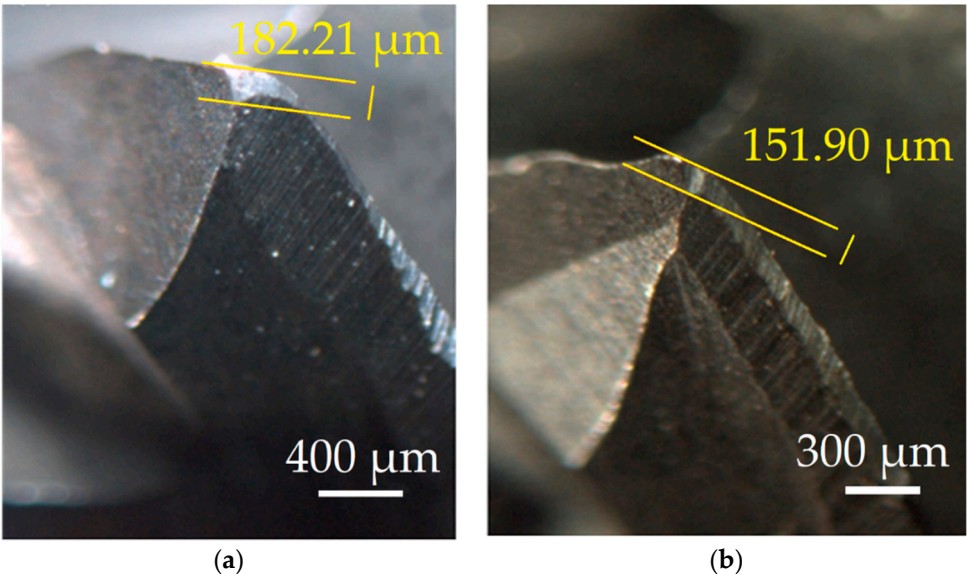

| (**a**) | (**b**) |

**Figure 14.** The size of the flank wear chamfer of end mills after 17.5 min of work when processing structural steel 41Cr4 with end mills; (**a**) uncoated, 182.21 μm; and (**b**) with a single-layer DLC, 151.90 μm.

Analysis of the development of wear in end mills after applying the sublayer (CrAlSi)N in combination with the formation of a DLC shows a noticeable effect of a two-layer coating on the tool wear rate when milling 41Cr4 structural steel. The tool life of end mills with (CrAlSi)N/DLC was 64 min, which is 2.0 times higher than the corresponding indicator of the uncoated tool (32 min) and 1.8 times higher than the tool with a single-layer DLC (35 min).

The dependences of the change in the surface roughness of the 41Cr4 steel workpiece on the operating time of the tool with different coatings shown in Figure 15 allow us to reveal the following regularity. From the very beginning of the cutting process, up to 25 min of work, the quantitative value of the roughness formed on the workpieces has a comparable value for all investigated mills and is in the range of 2.5–3.0 μm. After 25 min of operation, uncoated end mills show a noticeable deterioration in the quality of the machined surface and its sharp increase; at the time of tool failure (32 min),

the roughness of the workpiece is more than 6 μm. For mills with a single-layer DLC, a significant increase in the roughness of the processed workpiece is also observed, and by the time of tool failure (35 min), it is about 5 μm. The (CrAlSi)N/DLC makes a slightly more significant contribution to the reduction of roughness. Thanks to this coating, the surface roughness gradually increases over time, reaching the value of 8.75 μm at the time of failure (64 min).

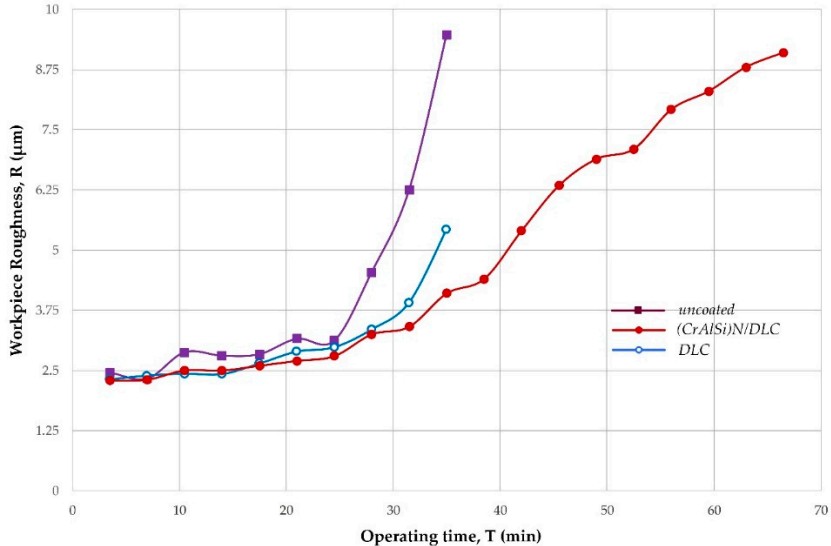

**Figure 15.** Dependence of the roughness of the treated workpiece surface of structural steel 41Cr4 on the operating time of end mills with different coatings.

## 4. Discussion of Research Results

The results of the XPS analysis allow us to judge the chemical composition and electronic state of atoms on the surface of the formed DLCs. From the experimental data obtained in Tables 4 and 5, it can be seen that DLCs of the two types studied (without sublayer and with sublayer (CrAlSi)N) have a very close ratio of components and approximately 64–66% consist of carbon atoms with different hybridization of valence electrons: $sp^3$-hybridization (diamond structure) and $sp^2$-hybridization (graphite structure). At the same time, the proportion of diamond hybridization (which is of particular interest to researchers) is almost identical for a single-layer DLC and a double-layer (CrAlSi)N/DLC. It is 47% and 51%, respectively, and the $sp^3/sp^2$ ratios of these coatings are similar at 3.0 and 3.4, respectively. First, these indicators show good reproducibility of PECVD technology used in work for the deposition of DLCs on a hard alloy. We add that this technology in a previous work was used by the authors for the deposition of DLC films on oxide-carbide ceramics [26,62,63], and the value of the $sp^3/sp^2$ ratio was close to that which was established when coating a hard alloy. Secondly, taking into account the fact that the volume fractions of the revealed silicon compounds on the surface of the DLC and (CrAlSi)N/DLCs were very close, which, taking into account the analysis error, allows us to conclude that the physicochemical processes occurring under the temperature conditions of PECVD technology do not lead to intensive chemical adsorption, mixing, and the formation of new phases in the formed DLC film. This is also indicated by the absence of compounds based on tungsten (the main component of the hard alloy), chromium, and aluminum (components of the CrAlSi sublayer) on the surface of DLCs, and the presence of silicon compounds in the two coatings are the result of discharge destruction of tetramethylsilane, which is a component of the gas mixture. It is necessary to clarify, speaking about the absence of chemisorption, we mean volumetric interaction processes inside the coating layer, while understanding that there is a chemical interaction at the substrate/coating interface, since it would be impossible to ensure good adhesive bond strength in its absence.

Thus, in reality, the structure (chemical composition and electronic state of atoms) of the formed DLC does not depend on the chemical composition and properties of the substrate on which it

is deposited but is determined by the strategy and parameters of the implementation of PECVD technology. At the same time, studies have shown that the physical and mechanical characteristics and strength of the adhesive bond of the formed DLC, on the contrary, strongly depend on the properties of the substrate on which they are deposited. Therefore, DLC films that are very similar in chemical composition and electronic state of atoms can have very different values of physical and mechanical characteristics, particularly the modulus of elasticity (Table 6).

The Martens hardness values measured by nanoindentation were very similar (Table 6) for all samples with DLCs (without sublayer and with sublayer (CrAlSi)N). At a load of 1.0 mN, the samples had hardness 33 and 34 GPa, and with increasing loads up to 4.0 mN and, respectively, with the increase in the contact depth, the hardness of samples decreased to 25 and 26 GPa. Interesting regularities were established concerning the quantitative values of the modulus of elasticity of hard alloy samples with different coatings. The data on the modulus of elasticity (*E*) of samples calculated during nanoindentation and shown in Table 6 essentially characterize the degree of rigidity of the surface layer of the tool material and its ability to deform elastically under external loading. We know that the lower the modulus of elasticity, the less rigid the material is with low deformation resistance, but it is less brittle and more ductile. If at a low load (1.0 mN), the *E* values for all samples were similar, 292 GPa for the DLC and 285 GPa for (CrAlSi)N/DLC, then at a higher load (4.0 mN), significant differences were found, 242 and 201 GPa. Analysis of the obtained data suggests that it is possible to identify areas in which there is a different nature of changes in the properties of the coating. For example, when an indenter is inserted to a depth of 90–100 nm, the properties of DLCs are determined by the surface characteristics. If the indenter is embedded to a depth of 380 nm or more, the properties of DLCs largely depend on the state and properties of the substrate and, of course, on the stresses at the "substrate-coating" interface. Hardness and modulus of elasticity cannot be considered separately, since they are interrelated, and their *H/E* ratio carries information about an important characteristic—the index of plasticity. A high value of the index of plasticity provides an increased service life in conditions of cyclic loads typical for milling, and the closeness of the values of the modulus of elasticity of the coating and the substrate helps to reduce technological stresses on the interface and increase the adhesive strength. Authoritative studies show that the ratio *H/E* = 0.15 characterizes the "ideal elasticity" of the coating [64–66]. In the present study, this indicator at a load of 1.0 mN was similar for DLC and (CrAlSi)N/DLCs, 0.113 and 0.119, respectively. In tests with a fourfold increase in load, the influence of the properties of the substrate with a preformed sublayer (CrAlSi)N was immediately indicated, the index of plasticity was almost 0.13, while for a single-layer DLC, this indicator was at the level of 0.1.

It seems logical to explain that the higher plasticity of the surface layer (coating) of the samples and the less stressed state at the "substrate-coating" interface had a decisive effect on the subsequently established higher adhesive bond strength of the DLC deposited on a carbide substrate with a pre-formed sublayer (CrAlSi)N. Of course, the hardness of the formed coating is a primary characteristic, but it is the elastic characteristics of the substrate and coating that have the most significant influence on the level of occurring stresses [67–70]. As a confirmation of this, we can consider the empirical dependence of the stresses in the coating, which appear due to the loading of the substrate, when tangential stresses occur in the plane of the adhesive contact, which can cause adhesive destruction of the coating. The maximum tangential stresses ($\tau_{max}$) in the plane of the adhesive contact are equal to [28]:

$$\tau_{max} = P\frac{L}{Ek} \tag{2}$$

where *P* is the load applied to the substrate (Pa); *E* is the modulus of elasticity of the substrate (GPa); *L* (Pa/m) and *k* (m$^{-1}$) are coefficients that depend on the modulus of elasticity of the coating.

It can be seen that by reducing the modulus of elasticity of the coating, it is possible to reduce the values of the maximum tangential stresses, the effect of which leads to the delamination of the coating, making it possible to increase its resistance to acting loads.



It is no coincidence that the experimental data shown in Figure 7 demonstrated that peeling of local areas of the coating from the substrate and especially the separation of the coating from a sample with a single-layer DLC, which has a higher modulus of elasticity, occurs at a significantly (almost two times) lower average force, at 12 N, while for the sample with (CrAlSi)N/DLC, which has a lower modulus of elasticity, at a load of 23 N.

The results of a comparative qualitative assessment of the adhesive bond strength of various coatings with hard alloy substrates once again confirm the above (Figure 8). Since local elastic-plastic deformation occurs around the indenter print during indentation, which causes swelling and destruction of the coating, it is the plasticity of the coating, which directly depends on the modulus of elasticity that is of paramount importance. A less plastic single-layer DLC at a uniform distance from the hole formed by the indenter has local areas of peeling of the coating and micro-chips, while a more plastic (CrAlSi)N/DLC after the load is removed, restores the surface layer, and there are no irreversible changes in it.

Since the resistance of the coating to abrasive wear is an important indicator that largely determines the performance of a coated product during operation, it is also necessary to discuss the results presented in Figure 9. The obtained images make it possible to approximate the surface area worn by external abrasive action, which can approximate the wear intensity. Unexpected results were not obtained here; the wear rate of hard alloy (CrAlSi)N/DLC-coated samples is 2.4 times lower than the uncoated samples and 1.5 times lower than the wear of single-layer DLC-coated samples.

Given that the hardness, which is the leading property that provides resistance to abrasion, two different coatings had very similar values (Table 6), such results can be explained as follows. First, it is the higher adhesive bond strength of the (CrAlSi)N/DLC with the substrate already discussed above. Secondly, it is necessary to consider the fact that after the external DLC layer is worn off; the nitride sublayer continues to function for some time and performs protective functions. Furthermore, the total thickness of (CrAlSi)N/DLCs is 4 µm, and the thickness of a single-layer DLC is 2.6 µm, which also affects the intensity of abrasive wear.

All of the above experimental results are very indicative and explain a lot, but they cannot consider the whole complex of actual operating loads. Therefore, it is possible to conclude the effectiveness of a particular coating only after discussing the results of operational tests of end mills with two types of DLCs when machining aluminum alloy and structural steel. In the operational tests carried out, the coatings showed significant differences in efficiency. The fact that when processing the aluminum alloy AlCuMg2, the application of a single-layer DLC significantly increased the durability of the end mill, and the application of a sublayer (CrAlSi)N did not contribute to an additional increase in the durability of the DLC-coated tool (Figure 10) can be explained as follows. The main task of the coating when processing aluminum-based alloys is to reduce the friction interaction on the contact pads of the cutter since the tool wear mechanism is associated with active cyclic adhesion of aluminum particles. The separation of particles from the contact surface of the tool and the workpiece is accompanied by the removal of micro-volumes of tool material from the working surfaces of the cutter, which are carried away from the cutting area by descending chips. However, a significant proportion sticks to the opposite surface of the part, reducing the quality of the treated surface. Therefore, due to the external DLC, which has excellent anti-friction characteristics, there is a decrease in friction interaction on the contact pads of the mill, and the development of the wear chamfer on the flank surface is significantly slowed down (Figures 10 and 11). The power loads when milling aluminum alloy are such that the coating on the cutting tool works in relatively favorable conditions and, according to experimental data (Figure 10), there is no need to form a sublayer under the external DLC (CrAlSi)N, thereby affecting the resistance of the coating to elastic and plastic deformations.

Analysis of the experimental data (Figure 13) obtained during the processing of 41Cr4 structural steel allows us to draw conclusions that differ significantly from those made above. This is not surprising, given that the maximum values of cutting forces that occur when processing structural steels can be four times higher than the corresponding values of force parameters when processing

aluminum alloys. Judging by the experimentally obtained dependences of tool wear on operating time (Figures 14 and 15), the external DLC quickly ceases to perform protective functions, and the conditions of contact interaction on the front and flank surfaces of the tool are similar to those that occur when using end mills without coatings. The role of the (CrAlSi)N sublayer under the influence of heat and power loads when milling 41Cr4 steel is manifested in the fact that more favorable conditions are created for the functioning of the external DLC, its adhesive bond strength with the tool material increases, the stress level in the coating decreases and it is capable of resisting the current heat and power loads for a longer time. Even after the abrasion of the outer DLC layer, the nitride sublayer can act as a wear-resistant coating, slowing down the wear rate. A consequence of the described processes is an increase in the resistance of the (CrAlSi)N/DLC to the applied loads and providing a twofold increase in the wear resistance of end mills compared to an uncoated tool when milling 41Cr4 steel (Figure 13) and 1.8 times compared to milling cutters with a single layer DLC.

The experimentally obtained dependences of the roughness of the machined surfaces of workpieces made of aluminum alloy AlCuMg2 and structural steel 41Cr4 on the operating time of end mills with various DLCs, presented in Figures 12 and 15, are in good agreement with the characteristic curves of the development of mill wear in time discussed above (Figures 10 and 13) and obey the classical principles of the theory of metal cutting. The wear of the cutting tool leads to an increase in the roughness of the machined surface because there is an increase in the actual contact area of the cutting tool and the workpiece being processed, and the intensity of their frictional and adhesive interaction increases. The use of DLCs hinders the development of these processes (Figure 12), thereby significantly reducing the average height of microroughness of the surface layer of an aluminum alloy part, which, even at the moment of tool failure (i.e., with a heavily worn tool), did not exceed 2.5 μm. In contrast to the milling of aluminum alloys during the forming of structural steel 41Cr4, the contribution of DLCs is significantly lower in reducing the intensity of friction and adhesive interaction between the tool and the workpiece. Therefore the effect on the roughness of the treated surface is not so pronounced. Nevertheless, the coating (CrAlSi)N/DLC slightly reduces the average height of the micro-roughness of the surface layer of the structural steel part (Figure 15), which is 2.5 μm in the first minutes of tool operation, and 8.75 μm at the time of tool failure.

## 5. Conclusions

The complex of experimental studies allowed us to obtain some actual results that can be used for further development of research in the field of development of DLC for the needs of tool production.

- The volume fraction of diamond hybridization in DLCs formed by PECVD technology in a gas medium containing tetramethylsilane is about 50%. It does not depend on the properties and chemical composition of the substrate, since the physical and chemical processes do not lead to intensive chemical adsorption and the formation of new phases in the formed DLC film. The chemical composition and electronic state of the DLC atoms depend on the strategy and parameters of the PECVD process.
- The hardness and modulus of elasticity of DLCs, on the contrary, are very strongly dependent on the properties of the substrate on which they are deposited. DLC films that are very close in chemical composition and electronic state of atoms can have very different values of physical and mechanical characteristics, particularly, the modulus of elasticity. When nanoindenting with a load of 4.0 mN, the positive effect of the properties of the substrate with a preformed sublayer (CrAlSi)N is sharply manifested; the index of plasticity (ratio of hardness and modulus of elasticity) for the (CrAlSi)N/DLC was almost 0.13, while for a single-layer DLC, this indicator was at the level of 0.1. The formation of the (CrAlSi)N intermediate sublayer has a significant effect on the ability of the DLC to resist elastoplastic deformations.
- Higher plasticity of (CrAlSi)N/DLCs and a less stressed state at the substrate-coating interface had an essential effect on increasing the adhesive bond strength of the coating to the substrate compared to a single-layer DLC. The separation of the coating from the sample with a single-layer

DLC, which has lower plasticity (greater modulus of elasticity), occurs at a significantly (almost two times) lower load. Simultaneously, the wear rate under the influence of abrasive particles on (CrAlSi)N/DLC-coated hard alloy samples are 2.4 times lower than uncoated samples and 1.5 times lower than single-layer DLC-coated samples.

- When milling aluminum alloy AlCuMg2, applying a single layer DLC increases the tool life of end mills many times (3.45 times), while applying a sublayer (CrAlSi)N does not contribute to an additional increase in the durability of a DLC-coated tool. This is due to the low thermal and power loads when machining aluminum-based alloys and the need to reduce frictional interaction on the contact pads of the cutter, which is entirely handled by a single-layer DLC.

- When milling 41Cr4 structural steel, the single-layer DLC very quickly ceases to perform protective functions and the contact conditions on the front and flank surfaces of the tool approach those that occur when using end mills without coatings. The application of (CrAlSi)N/DLC alone provided an increase in the durability of end mills up to two times. The role of the (CrAlSi)N sublayer in milling steel 41Cr4 was manifested in providing more favorable conditions for the functioning of the external DLC, increasing its adhesive bond strength to the substrate, reducing the stress level in the coating, and, as a result, the (CrAlSi)N/DLC was able to resist the existing heat and power loads a longer time.

- The effect of DLCs on the roughness of the machined surface obeys the classical principles of the theory of metal cutting—an increase in the wear of the cutting tool leads to an increase in the roughness of the machined surface. The use of single-layer DLCs noticeably reduces the average height of microroughness of the surface layer of a workpiece made of aluminum alloy AlCuMg2, which, even at the moment of cutter failure, does not exceed 2.5 µm. In contrast to the milling of aluminum alloys, when forming structural steel 41Cr4, the role of DLCs in reducing the intensity of the frictional and adhesive interaction of the tool and the workpiece being processed is not so pronounced, and, as a consequence, in reducing the roughness of the processed surface. However, the (CrAlSi)N/DLC has a definite effect on reducing the average roughness of the surface layer of a structural steel part.

**Author Contributions:** Conceptualization, S.N.G.; methodology, M.A.V.; validation, S.V.F.; formal analysis, M.A.V. and S.V.F.; investigation, M.M. and S.V.F.; resources, S.N.G.; data curation, M.A.V.; writing—original draft preparation, S.V.F. and M.A.V.; writing—review and editing, S.N.G., M.A.V., S.V.F., and M.M.; visualization, M.M.; supervision, M.A.V.; project administration, S.N.G. All authors have read and agreed to the published version of the manuscript.

**Funding:** This work was supported by the Russian Science Foundation under grant 18-19-00599.

**Acknowledgments:** The work was carried out using the equipment of the Center of collective use of MSUT STANKIN.

**Conflicts of Interest:** The authors declare no conflict of interest.

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
