# Peer review of "Influence of DLC Coatings Deposited by PECVD Technology on the Wear Resistance of Carbide End Mills and Surface Roughness of AlCuMg2 and 41Cr4 Workpieces"

_coatings, doi:10.3390/coatings10111038_

Round 1
Reviewer 1 Report
Manuscript title: Influence of DLC coatings deposited by PECVD technology on the wear resistance of carbide end mills and surface roughness of AlCuMg2 and 41Cr4 workpieces
Authors: Sergey Grigoriev, Marina Volosova, Sergey Fyodorov and Mikhail Mosyanov
Manuscript No: coatings-961738
The manuscript deals with wear resistant coatings (specifically DLC and (CrAlSi)N/DLC) prepared by PECVD deposition. The manuscript gives performance evaluation of the coatings and interprets the obtained data. The study is of interest due to thorough description of various technical details. There are still some errors in the text and some figures need an improvement:
1. Typos and grammar errors:
In several places a gap between two words are missing (e.g. line 134 The choice of (CrAlSi)Nas an adhesive..., line 236 Kαand Al Kαradiation…, )
2. Line 551 ...chips, while a more plastic (CrAlSi)N/DLC coatings after the load is removed, restores the surface layer is, ... “is” can be left out
3. Line 670 S.V.- there is no author’s name with acronym S.V. in the author list.
4. Line 415 …The experimental data obtained during the tests (Figure 11) shows that when… “show” is correct.
5. Table 6: HPa is likely wrong unit (GPa may be right)
6. Line 165 ...centre by DMG... The meaning of the abbreviation (DMG) is not given in the text.
7. 5b, the label in the figure is vague.
8. Figs 10 and 13: the color of the lines and writings (red) is not suitable (bad contrast), yellow color would be better.
9. Figs 12 and 14. It seems that DLC and (CrAlSi)N/DLC lines are interchanged (see text … line 442 the tool wear rate when milling 41Cr4 structural steel. The tool life of end mills with (CrAlSi)N/DLC coating was 64 minutes,…)
10. Line 537 …It can be seen that by reducing the modulus of elasticity of the coating, it is possible to reduce the values of the maximum tangential stresses...
But according to formula 2, the reduction of E results in an increase of tau_max which seems to be in contradiction with the above mentioned statement from the authors.
Author Response
Response to Reviewer 1 Comments
Dear reviewer,
Thank you very much for your kind evaluation of our work. We do agree with all your proposals and comments and have modified the manuscript according to them.
Introduced changes were marked by yellow in the text of the manuscript.
Kind regards,
Authors.
1. Typos and grammar errors: In several places a gap between two words are missing (e.g. line 134 The choice of (CrAlSi)Nas an adhesive..., line 236 Kαand Al Kαradiation…, )
Thank you very much for your kind remark. It is a problem with incorrect work of Word software; we have tried to revise it where it was possible.
2. Line 551 ...chips, while a more plastic (CrAlSi)N/DLC coatings after the load is removed, restores the surface layer is, ... “is” can be left out
Thank you, it is revised.
3. Line 670 S.V.- there is no author’s name with acronym S.V. in the author list.
Thank you, there is Dr. S.V. Fedorov. “V.” is for a patronymic name. The full name is Dr. Sergey Voldemaromich Fedorov, in the normal life we call him respectfully “Sergey Voldemaromich”. We have revised the names of older colleagues by mentioning the first letters of their patronymic names to make it easier later to identify publication in the international citation systems since there are two Sergeys Fedorovs in our university and in common more than 10 in all over the world.
4. Line 415 …The experimental data obtained during the tests (Figure 11) shows that when… “show” is correct.
Thank you, it is revised.
5. Table 6: HPa is likely wrong unit (GPa may be right)
Thank you, it is revised.
6. Line 165 ...centre by DMG... The meaning of the abbreviation (DMG) is not given in the text.
DMG is for DMG MORI CO., LTD. It is revised in the text.
7. 5b, the label in the figure is vague.
Thank you, it is revised.
8. Figs 10 and 13: the color of the lines and writings (red) is not suitable (bad contrast), yellow color would be better.
These figures have an informative character when the red color is standard for this microscope type. We can revise them although we did not find a point in it or delete them totally from the manuscript if the reviewer finds the red color of lines and data critical. The relevant data on flank wear is added to the figures’ titles.
9. Figs 12 and 14. It seems that DLC and (CrAlSi)N/DLC lines are interchanged (see text … line 442 the tool wear rate when milling 41Cr4 structural steel. The tool life of end mills with (CrAlSi)N/DLC coating was 64 minutes,…)
Thank you, we have noticed it as well. That was a technical mistake. The figures are revised.
10. Line 537 …It can be seen that by reducing the modulus of elasticity of the coating, it is possible to reduce the values of the maximum tangential stresses...
But according to formula 2, the reduction of E results in an increase of tau_max which seems to be in contradiction with the above mentioned statement from the authors.
10. Thank you very much. Probably it is not evident from the provided formula, where E is Young’s modulus for the substrate. The relevant explanation is added in the file and provided in the text of the manuscript.
Reviewer 2 Report
Review of
Influence of DLC….workpieces
By
Grigoriev, et. al..
Summary:
The authors put PACVD DLC on carbide end mills, with and without a bonding layer, and machine an Al alloy and an iron alloy. Flank wear and workpiece roughness are reported. Some other characterization is done.
Evaluation, necessary corrections, and comments
1. This work does not meet the minimum criteria for publishable work. Decades ago it would have been acceptable to report simple DLC results because all findings were new. Now, not only has much research been reported, but the research has been commercialized end mills are available with many coatings including DLC. The amount of work done here far insufficient for any publication.
2. The authors must eliminate many paragraphs, because these paragraphs are essentially off-topic essays. The reviewer has seen this type of thing before: where authors write lengthy text which has nothing to do with the subject of the research being done. The discussion on residual stress is an example of this. The authors do not quantify residual stress, they do not study thickness effect on residual stress, they do not study voltage effect on residual stress, they do nothing on residual stress. Thus the section on residual stress must be eliminated. A good research paper dwells primarily on the new findings, not on ancillary subjects covered by a school textbook.
3. The authors seem to be new to the field and unaware of what is going on, and thus write at length about things which are obvious to any researcher. The reviewer finds himself saying things like “it is obvious that the bond layer helps the DLC to stick to the surface, and it is obvious and expected that the Rockwell circle is smaller, and it is obvious and expected that the critical load is larger.”
4. Fig 14 is extraordinary because it shows that the machining properties depend adversely on the bonding layer, which is possible. Because of this, it is necessary for the authors to examine and provide a micrograph of the cutting edges for the reader so that the reader can attempt to understand why this counterintuitive behavior is occurring.
5. Much of the “discussion” is verbiage without connection the paper. Indeed, almost all of the discussion, all of the #500 lines, can be eliminated because they say nothing. They read like a chapter from a school textbook. A technical article is not a textbook.
6. The reviewer wonders if the authors know much about the plasticity index, as DLC is rather brittle and when the load is large, it is expected that more of the bonding layer contributes to the curve measured by the nanoindenter.
7. Most of the conclusions aren’t conclusions. Instead they tend to be simply a restatement of the experimental findings. Conclusion 2 is not a conclusion and suggests that the authors don’t know hardness testing. It is understandable that the numbers at 4 mN are smaller than at 1 mN because the tip is probing a deeper and softer layer. #2 is actually wrong. Table 6 shows the hardness and modulus of elasticity do not depend on the substrate. To within error, 33=34, and 292=285
8. Conclusion 3 makes assertions which are not supported. The authors did no work to determine the stress state.
9. Conclusion 4 makes assertions which are not warranted. The entire text about friction and power loads is an unsupported assumption.
10. Conclusion 5 says for steel, the plain DLC is not very helpful in machining, but the bonding layer helps a lot. Figs. 12 and 14 say the opposite, that the plain DLC is very good for wear and smoothness.
11. Nowhere do the authors state whether the machining was performed dry or with a fluid. This information must be put into the abstract at the beginning so the potential reader will know if it is pertinent to them or not.
12. The author’s citation list is very large. Because of this they must know that hard coatings on machine tools is a very well researched topic and the technology is now not only experimental, but commercially available. DLC coatings on end mills are now so common that the reviewer can buy them from a local supply house. The supply house has DLC coated carbide end mills, diamond coated carbide end mills, TiN coated carbide end mills, multi-layer TiN/TiAlN coated carbide end mills, TiCN coated carbide end mills, and plain carbide end mills. Thus the authors must produce a more insightful treatment, instead of using only one kind of DLC, at one thickness, of one composition, at one voltage, at one feed rate, at one speed, for their research. It is quite proper to perform experiments completely with the P1311 unit, but at different thicknesses, or different voltages, or different bond layers, or with different turning speeds. If the reader can make no comparisons then nothing is learned.
13. Figure 9 is not surprising. Figure 12 is extremely surprising. It is surprising because the better bonded coating performs worse, and also it shows that the cutting performance “sees” what is underneath the DLC coating for steel, but not for Al. This is interesting. Can the authors remark upon this? Is the coating still present or has it been accidently removed?
14. The authors need to specify the deposition pressure because this is a fundamental parameter and has an effect on hardness.
15. The films themselves are more properly called a-H:C. or perhaps a-C:Si:H, because they are amorphous and they contain much hydrogen and some Si.
16. Fig. 10 is of poor quality. Fig. 13 is a little better.
17. Most of the introduction must be eliminated. For example, lines 27-42 can be eliminated. Everyone knows that machining and end mills are important. This reads like a popular trade journal, not a research article.
18. Fig. 1 and most of the discussion must be eliminated. The authors are nowhere doing research on DLC, and thus this has nothing to do with the manuscript. The authors are using a single type of DLC and studying flank wear. The workers must concentrate on their findings, and not attempt to teach the basics of DLC structure, which is irrelevant to the paper. On the other hand, if the workers had performed a verity of experiments with different feed gases and targets then this would have been a DLC paper, but they did not do that.
19. The authors must detail how flank wear is measured. There seems to be only a few lines of explanation. This is important. Is it the width? The volume? How large an area? Is it proportional to any physical quantity? Is it uniform? Do the authors measure at a specific distance from the cutting edge?
20. An image should never be referred to as having a certain magnification, for example line 299, because the final printed size is not known. Instead, every image must have a scale bar on it.
21. The reviewer commends the authors for the detailed XPS and flank wear data obtained at frequent intervals. The shape of some of the curves loo similar, almost parabolic. Can the shape be related to a fundamental wear mechanism? Can the shape say anything about carbide vs. coated?
22. 50% sp3 is large. A reader might wonder what kind of DLC is being made. What does the literature say?

Author Response
Response to Reviewer 2 Comments
Dear reviewer,
Thank you very much for your kind evaluation of our work. We do agree with all your proposals and comments and have modified the manuscript according to them.
Introduced changes were marked by green in the text of the manuscript.
Kind regards,
Authors.
1: This work does not meet the minimum criteria for publishable work. Decades ago it would have been acceptable to report simple DLC results because all findings were new. Now, not only has much research been reported, but the research has been commercialized end mills are available with many coatings including DLC. The amount of work done here far insufficient for any publication.
Thank you very much for your kind evaluation of our work. We have tried to improve our manuscript according your comments and hope that our highlighted novelty will be at the level of the Journal’s requirements.
The novelty is .that we have shown the role of the underlayer (indicate our composition) in the composition of the DLC film obtained by the CVD process during the decomposition of the gas mixture into the basic physical and mechanical characteristics of the coating, the wear resistance of the tool for various types of processed material, i.e. under various operating loads, the aim of this work was to comprehensively study the effect of the formation of an adhesion underlayer based on (CrAlSi) N on carbide end mills before applying an external DLC film using the PECVD method in the presence of a multicomponent gas mixture containing tetramethylsilane, in comparison with the application of only one layer DLC coatings and establish the degree of influence of the adhesive sublayer on the important physical, mechanical and structural characteristics of the DLC coating.
2. The authors must eliminate many paragraphs, because these paragraphs are essentially off-topic essays. The reviewer has seen this type of thing before: where authors write lengthy text which has nothing to do with the subject of the research being done. The discussion on residual stress is an example of this. The authors do not quantify residual stress, they do not study thickness effect on residual stress, they do not study voltage effect on residual stress, they do nothing on residual stress. Thus the section on residual stress must be eliminated. A good research paper dwells primarily on the new findings, not on ancillary subjects covered by a school textbook.
We would like to introduce our discussion on the influence of coating thickness, elasticity modulus, and load on shear stress with quantitative evaluation of τmax. The detailed discussion on residual stresses is provided in Point 8. Let us see more detailed the known formula of maximal shear stress (a detailed explanation is given in the file).
When cutting, the acting stresses are associated with the modes, cutting forces, and the length of contact of the chips with the front surface, which depends on the coefficient of friction.
With an increase in Young's modulus of the coating, the tangential stress increases by inverse square; with an increase in Young's modulus of the substrate, the tangential stress decreases by inverse square. With an increase in the coating thickness, the tangential stress increases by inverse square; with an increase in the thickness of the substrate, the tangential stress decreases by inverse square.
For the first load, with a decrease of 2.4% in elasticity modulus, the reduces from 97.15 to 90.80 MPa; for the second load, with a decrease of 16.9% in elasticity modulus, the will be reduced from 18.90 to 17.23 MPa.
At the same time, if E modulus stays constant and average (267 and 243 GPa), with a decrease of 9% in elasticity modulus, the will be reduced only by ~5%.
It should be noted that used for calculating the theoretical E modulus and Poisson’s ratio for substrate were similar for all type coatings when in practice E modulus will vary and the effect of doubled coating will be obviously greater.
Thus, for the first load, with a decrease of 2.4% in elasticity modulus, the will be reduced by from 113.95 to 112.6 MPa; for the second load, with a decrease of 16.9% in elasticity modulus, the will be reduced by from 23.45 to 21.37 MPa. With the greater load, the effect is more evident.
With an increase in coating thickness by ~1.5 times, the stress grows by ~15-19%.
Table 2 (of response). Calculated shear stress , MPa
|
Indenter load, mN |
Shear stress , MPa |
|
|
Single coating |
Doubled coating |
|
|
Thickness of 2.6 µm |
||
|
1 |
97.15 |
90.80 |
|
4 |
18.90 |
17.23 |
|
Thickness of 4.0 µm |
||
|
1 |
113.95 |
112.60 |
|
4 |
23.45 |
21.37 |
We should note that the Poisson’s ratio is from 0.05 to 0.25 for DLC coatings deposed at -500V and 26 Pa and from 0.15 to 0.25 for DLC coatings deposed at -800V and 13 Pa, then we can see that even for single-mode the Poisson’s ratio can vary in the wide range of values [2]. The average value of the Poisson’s ratio for DLC coatings equal to 0.12÷0.18.
It should be noted that the Poisson’s ratio for most of the materials is in the range of 0÷0.5 then it influences E value in a range of 5% [3].
The relevant explanation is provided in the text of the manuscript.
If the reviewer will point us the paragraphs to be eliminated, we will satisfy this requirement.
References
- Dolgov, N.A. Analytical Methods to Determine the Stress State in the Substrate-Coating System Under Mechanical Loads. Strength Mater. 2016, 48, 658-667.
- Bec, S.; Tonck, A.; Fontaine, J. Nanoindentation and nanofriction on DLC films. Philosophical Magazine 2006, 86, 33-35, 5465-5476.
- Topolyansky, P. A.; Sharifullin, S. N.; Adigamov, N. R.; et al. Finished plasma strengthening and restoration of fuel equipment details. Journal of Physics Conference Series2018, 1058, UNSP 012075.
3. The authors seem to be new to the field and unaware of what is going on, and thus write at length about things which are obvious to any researcher. The reviewer finds himself saying things like “it is obvious that the bond layer helps the DLC to stick to the surface, and it is obvious and expected that the Rockwell circle is smaller, and it is obvious and expected that the critical load is larger.”
We would like to confirm that our research team is quite solid and has experience in researching coatings for more than 30 years with more than 200 papers related to the deposition of thin coatings. Some of our most cited works are:
- Vereschaka, Alexey A.; Grigoriev, Sergey N. Study of cracking mechanisms in multi-layered composite nano-structured coatings. WEAR 2017, 378-379, 43-57.
- Vereshchaka, A. A.; Vereshchaka, A. S.; Mgaloblishvili, O.; et al. Nano-scale multilayered-composite coatings for the cutting tools. INTERNATIONAL JOURNAL OF ADVANCED MANUFACTURING TECHNOLOGY 2014, 72, 1-4, 303-317.
- Vereschaka, A. A.; Grigoriev, S. N.; Sitnikov, N. N.; et al. Delamination and longitudinal cracking in multi-layered composite nanostructured coatings and their influence on cutting tool life. WEAR, 2017, 390-391, 209-219.
- Volkhonskii, A. O.; Vereshchaka, A. A.; Blinkov, I. V.; et al. Filtered cathodic vacuum Arc deposition of nano-layered composite coatings for machining hard-to-cut materials. INTERNATIONAL JOURNAL OF ADVANCED MANUFACTURING TECHNOLOGY 2016,84, 5-8, 1647-1660.
- Sobol', O. V.; Andreev, A. A.; Grigoriev, S. N.; et al. Effect of high-voltage pulses on the structure and properties of titanium nitride vacuum-arc coatings. METAL SCIENCE AND HEAT TREATMENT 2012, 54, 3-4, 195-203.
- Grigoriev, Sergej N.; Vereschaka, A. A.; Vereschaka, A. S.; et al. Cutting tools made of layered composite ceramics with nano-scale multilayered coatings. Procedia CIRP 2012, 1, 301-306.
- Sobol', O. V.; Andreev, A. A.; Grigoriev, S. N.; et al. Vacuum-arc multilayer nanostructured TiN/Ti coatings: structure, stress state, properties. METAL SCIENCE AND HEAT TREATMENT 2012, 54, 1-2, 28-33.
- Fominski, V. Yu.; Grigoriev, S. N.; Celis, J. P.; et al. Structure and mechanical properties of W-Se-C/diamond-like carbon and W-Se/diamond-like carbon bi-layer coatings prepared by pulsed laser deposition. THIN SOLID FILMS 2012, 520, 21, 6476-6483.
- Vereschaka, A. A. .; Volosova, M. A.; Batako, A. D.; et al. Development of wear-resistant coatings compounds for high-speed steel tool using a combined cathodic vacuum arc deposition. INTERNATIONAL JOURNAL OF ADVANCED MANUFACTURING TECHNOLOGY 2016, 84, 5-8, 1471-1482.
Short presentation of the research group:
|
Authors |
H-index in Web of Science |
ResearcherID Web of Science |
Published papers in the journals, included in Web of Science |
Editorship |
|
Prof., Dr. in Eng. Sci., Sergey N. Grigoriev |
26 |
AAF-8027-2019, J-2463-2012, AAF-8027-2019, |
205 |
Coatings, Metals (2019-2020), Mechanics and Industry (2015-2017), Material Science Forum (2015,2016) |
|
Doc., Dr. in Ph. (Eng. Sci.), Marina A. Volosova |
15 |
B-3020-2013 |
126 |
Metals (2019-2020), Mechanics and Industry (2015-2017), Material Science Forum (2015,2016) |
|
Dr. in Ph. (Eng. Sci.), Sergey V. Fedorov |
3 |
S-8840-2018 |
31 |
- |
|
Mr. Mikhail Mosyanov |
- |
- |
2 |
- |
We would like to note that open access journals are often used as data sources for students and young scientists with no access to the paid science source. We would like to note that the most authoritative journals are closed for access, and nowadays, science becomes a privilege for the student that studies in the most advanced higher schools that are available to pay access to the bases of famous publishing houses. We are for open science; it is why we choose for a part of our publication open access and ready to pay publication processing to make science accessible for the full range of potential readers, for everyone who wants to know more, to research more. However, due to the orientation to the partly inexperienced audience, we find it very important to pronounce the obvious things for us but should be pronounced once again to be very understandable for everyone. Moreover, there is nothing to do with the volume of the material we would like to publish since even 10-15 pages are enough for a publication in Nature, for example.
4. Fig 14 is extraordinary because it shows that the machining properties depend adversely on the bonding layer, which is possible. Because of this, it is necessary for the authors to examine and provide a micrograph of the cutting edges for the reader so that the reader can attempt to understand why this counterintuitive behavior is occurring.
That was a technical mistake. The figures are revised.
5. Much of the “discussion” is verbiage without connection the paper. Indeed, almost all of the discussion, all of the #500 lines, can be eliminated because they say nothing. They read like a chapter from a school textbook. A technical article is not a textbook.
Thank you for your kind suggestion. We would like to have mentioned should we remove the lines before Line #500 or after?
6. The reviewer wonders if the authors know much about the plasticity index, as DLC is rather brittle and when the load is large, it is expected that more of the bonding layer contributes to the curve measured by the nanoindenter.
And about the fact that the reviewer finds DLC are fragile - well, it’s not like that since the measured hardness was very modest - 36 GPa. This is not the hardness at which we can talk about excessive fragility but just the sublayer increases its plasticity, as can be judged from the table, lines 2 and 4. The relevant passage is added to the text.
7. Most of the conclusions aren’t conclusions. Instead they tend to be simply a restatement of the experimental findings. Conclusion 2 is not a conclusion and suggests that the authors don’t know hardness testing. It is understandable that the numbers at 4 mN are smaller than at 1 mN because the tip is probing a deeper and softer layer. #2 is actually wrong. Table 6 shows the hardness and modulus of elasticity do not depend on the substrate. To within error, 33=34, and 292=285
As indicated in work, when assessing the hardness and elastic modulus, we used the method of W. Oliver and G. Pharr. This technique has proven itself well, although, like any other instrumental method, has its specifics. The judgment that Table 6 shows that the hardness and the modulus of elasticity are independent of the substrate and that the measured values cannot form the basis for conclusions, in our opinion, is not justified. The indentation method proposed many years ago by W. Oliver and G. Pharr [1] was the first to substantiate the principles that allow, in contrast to other classical methods of measuring the hardness of films (for example, the Vickers pyramid), to carry out measurements in order to exclude the influence of the substrate and to provide an opportunity to evaluate precisely the properties of the coating (this occurs at low loads on the Berkovich indenter). At the same time, these authoritative scientists in their further works showed that with an increase in the load, the response of the indenter when it penetrates the sample is determined not only by the mechanical properties of the coating, but also by the properties of the substrate, and with an increase in the penetration depth, the contribution of the substrate increases and the deposition of a harder film on a softer substrate [2]. Another work of the classics - [3].
In modern devices for nanoindentation, the scientific principles laid down many years ago have been brought to an exceptionally high level and allow measurements to be made with high accuracy, particularly the Micro Scratch Tester by CSEM, used by us, used by the world's leading laboratories.
Existing work in this area, for example [4] show when measuring the hardness and elastic modulus for all "depth" curves, three intervals can be distinguished, in which a different character of changes in properties is observed: 1 - the interval within which the properties of the coating are determined by the state surfaces; 2 - interval within which the properties of the coating are relatively constant; 3 - the interval within which the properties of the coating are determined by the state and properties of the substrate, as well as by the stresses at the interface between the coating and the substrate.
These are general principles; they are specified and developed by research teams for a specific pair of "substrate - coatings". In our work, we received original data for our pairs "Hard alloy / DLC" and "Hard alloy / (CrAlSiN / DLC". We got that interval 1 - at the penetration of the indenter (indentation depth) up to 150 nm, interval 2 - at penetration indenter 150-350 nm, interval 3 - over 350 nm. This is indeed the case and the data in Table 6. It is clearly seen that at an indentation depth of 380 nm, the substrate already has a serious effect.
Table 6 (of the manuscript). Physical and mechanical characteristics of DLC coatings deposited on hard alloy samples
|
№ |
Type of test sample |
Applied load (mN) |
Penetration depth, nm |
Hardness HM (GPa) |
Modulus of elasticity E (HPa) |
Index of plasticity (H/E) |
|
1 |
Hard alloy /DLC |
1.0 |
90 |
33±3 |
292±8 |
0.113 |
|
2 |
Hard alloy/(CrAlSiN/DLC |
34±2 |
285±10 |
0.119 |
||
|
3 |
Hard alloy /DLC |
4.0 |
380 |
25±2 |
242±7 |
0.10 |
|
4 |
Hard alloy/(CrAlSiN/DLC |
26±3 |
201±8 |
0.129 |
As for the scatter when measuring the elastic modulus and hardness, the data obtained are traditional for this measurement method and the resulting errors are classic. See the latest articles in Coatings magazine as an example [5,6].
If necessary, numerous references from other authoritative publications can be cited, confirming that the results of these studies, based on nanoindentation, can be interpreted and appropriate conclusions can be drawn.
References:
- Oliver and G. Pharr. An improved technique for determining hardness and elastic modulus using load and displacement sensing indentation experiments. J. Mater. Res. 1992, 7 (6), 1564-1583.
- Y. Tsui, C.A. Ross, and G.M. Pharr, "Monoindentation Hardness of Soft Films on Hard Substrates: Effect of the Substrate," Mater. Res. Soc. Symp. Proc. 493, 57-62 (1997)
- Ranjana Saha, William D. Nix. Effects of the substrate on the determination of thin film mechanical properties by nanoindentation, Acta Materialia, 2002, 50 (1), 23-38.
- A. Levashov, M.I. Petrzhik, M.Ya. Tyurina (Bychkova), F.V. Kiryukhantsev-Korneev, P.A. Tsygankov, and A.S. Rogachev. Multilayer nanostructured heat-generating coatings. Preparation and certification of mechanical and tribological properties. Metallurgist, Vol. 54, no. 9-10, 2011, pp. 623-634.
- Dang, B.; Tian, T.; Yang, K.; Ding, F.; Li, F.; Wei, D.; Zhang, P. Wear and Deformation Performance of W/Ta Multilayer Coatings on Pure Cu Prepared by Double Glow Plasma Alloying Technique. Coatings2020, 10, 926. https://www.mdpi.com/2079-6412/10/10/926/htm
- Huang, D.; He, W.; Cao, X.; Jiao, Y. Investigations in Anti-Impact Performance of TiN Coatings Prepared by Filtered Cathodic Vacuum Arc Method under Different Substrate Temperatures. Coatings2020, 10, 840. https://www.mdpi.com/2079-6412/10/9/840
8. Conclusion 3 makes assertions which are not supported. The authors did no work to determine the stress state.
The authors of this work are undoubtedly familiar with reputable scientists' works on the study of the effect of application conditions on the stress state of the coating. These works and some of our other works speak about the interdependence of the adhesive bond's strength with the stress state. We considered that knowing the general laws, and we can draw particular conclusions. Namely, having determined improved adhesion, we associated this with lower residual stresses. Perhaps our mistake was that we did not additionally support these conclusions with experimental results. We were preparing for publication a separate study devoted to the change in residual stresses in the DLC coating; some of our results are true of interest. We did not plan to add them to this article, but considering the criticism that arose, we added separate experiments to study the values of residual stresses in our samples.
Section 2.3. was renamed
was: 2.3. X-ray photoelectron analysis of DLC coatings
now: 2.3. X-ray photoelectron and diffraction analysis of DLC coatings
At the end of section 2.3. we added a paragraph:
To study the stresses, the authors used a PANALYTICAL EMPYREAN X-ray diffractometer with monochromatic CuKα-radiation. The stresses were determined by the classical sin2y method with a grazing beam at a fixed angle of incidence of the beam y0 and scanning along the 2q axis.
Section 3.2. was renamed
was: 3.2. Microhardness, modulus of elasticity and adhesion strength of DLC coatings
now: 3.2. Microhardness, modulus of elasticity, residual stress and adhesion strength of DLC coatings
Immediately after Table 6, we added the results of the measurement of residual stresses and a new Figure 6 (we shifted the numbering of all subsequent figures by one).
The explanatory text was added:
As follows from the results of the analysis of samples on an X-ray diffractometer (Figure 6), the character of the distribution of residual compressive stresses in the coating for samples with a single-layer DLC coating and a (CrAlSi)N/DLC coating does not change significantly. Attention is drawn to a noticeable decrease in residual stresses' average values when a DLC coating is applied to a sample with a (CrAlSi)N sublayer. Average residual stresses for single-layer DLC coatings are 1250÷3650 MPa, while for DLC coatings with a sublayer - 650÷2750 MPa.
9. Conclusion 4 makes assertions which are not warranted. The entire text about friction and power loads is an unsupported assumption.
Our conclusion No. 4 was formulated as follows: When milling aluminium alloy AlCuMg2, applying a single layer DLC coating increases the tool life of end mills many times (3,45 times), while applying a sublayer (CrAlSi)N does not contribute to an additional increase in the durability of a DLC-coated tool. This is due to the low thermal and power loads when machining aluminium-based alloys and the need to reduce frictional interaction on the contact pads of the cutter, which is entirely handled by a single-layer DLC coating.
We considered that there is no need to provide additional (confirming) information since the fact that the heat and power loads during the processing of an aluminum alloy are much less than during the processing of steel is too obvious a fact. It is also known from numerous works about the positive effect of DLC coatings on reducing frictional interaction.
Different physical and mechanical characteristics of AlCuMg2 alloy and 41Cr4 steel (see table 3 data for milling with end mills with a diameter of 6 mm at a cutting speed of 200 m/min) cause dramatically different power loads and temperatures during milling.
Table 3 (of response). Physical and mechanical characteristics of AlCuMg2 alloy and 41Cr4 steel and it influence on power loads and temperatures during milling with end mills with a diameter of 6 mm at a cutting speed of 200 m/min.
|
Processed material |
Stress strength (MPa) |
Hardness |
Feed value (mm/tooth) |
Cutting force value Pz (N) |
Average cutting temperature T (оС) |
|
AlCuMg2 |
245 |
105 НВ |
0.015 |
1106 |
167 |
|
0.068
|
1651 |
194 |
|||
|
41Cr4 |
635 |
210 НВ |
0.015 |
2508 |
398 |
|
0.068
|
3107 |
426 |
These data are not unexpected, and they are calculated based on classical techniques outlined in many works [1-4]. Nevertheless, taking into account the existing comment of the reviewer, in subsection 2.1. Processed materials, cutting tools, and operational testing methods. Right after Table 2, we added a paragraph:
The processed materials used in work differ significantly in their physical and mechanical characteristics - the aluminum alloy AlCuMg2 has a tensile strength of 245 MPa with a hardness of 105 HB, and 41Cr4 steel - 635 MPa and 210 HB, respectively. Such differences predetermine a significant difference in the heat and power loads that act on the cutter's cutting edge during cutting. When machining 41Cr4 steel, more than a twofold increase in the component of the cutting force Pz and the temperature in the cutting zone is observed.
In addition, in order to eliminate the comments of the reviewer, we clarified subsection 2.5
WAS
2.5. Abrasion resistance of DLC coatings
NOW
2.5. Friction сoefficient and abrasion resistance of DLC coatings
In subsection 2.5, we added the first paragraph:
The friction coefficient was evaluated in work on a Tetra Basalt N2 testing machine from TETRA GmbH to assess the DLC coating's effect on the transformation of frictional properties. During the tests, the coefficient of friction-sliding of rubbing pairs "hard alloy with DLC-coating - a counter body made of AlCuMg2 alloy" and "hard alloy with DLC-coating - a counter body made of 41Cr4 steel" were determined. The tests of all samples were carried out under conditions of dry friction at identical normal loads on the counter body (1 N), with the speed of relative displacement being 2 mm·s−1 and the friction path being 90 mm. We evaluated the coefficient of friction for samples with single-layer DLC coatings.
To demonstrate the obtained results, we have specified subsection 3.3.
WAS
3.3. Abrasion resistance of DLC coatings
NOW
3.3. Friction сoefficient and abrasion resistance of DLC coatings
At the very beginning of the section, we added table 7 with the results and after it a text explanation
Table 7 (of the manuscript). The friction coefficient of DLC coatings deposited on hard alloy samples
|
No. |
Type of sample |
Type of counter body |
Friction coefficient value |
|
1 |
Hard alloy 6WH10F |
AlCuMg2 |
0.29÷0.32 |
|
2 |
Hard alloy 6WH10F/ DLC coating |
AlCuMg2 |
0.15÷0.16 |
|
3 |
Hard alloy 6WH10F |
41Cr4 |
0.41÷0.44 |
|
4 |
Hard alloy 6WH10F/ DLC coatings |
41Cr4 |
0.25÷0.27 |
As shown from table 7, DLC-coating significantly reduces the hard alloy's friction on the aluminum alloy - from 0.29÷0.32 to 0.15÷0.16. The effect of coating deposition in friction against steel is also noticeable - the coefficient of friction coefficient was reduced from 0.41÷0.44 to 0.25÷0.27.
References:
- Stability Limits of Milling Considering the Flexibility of the Workpiece International Journal of Machine Tools & Manufacture, 2005. 45(15), 1669–1680.
- A long-term control scheme of cutting forces to regulate tool life in end milling processes // Precision Engineering, 2010. 34 (4), 675–682.
- Prediction of regenerative chatter by modeling and analysis of high-speed milling. Int J Mach Tools Manuf, 2003. 43(14), 1437–1446.
- Modeling and measurement of cutting temperatures in milling. Proc. CIRP, 2016, 46, 173–176.
10. Conclusion 5 says for steel, the plain DLC is not very helpful in machining, but the bonding layer helps a lot. Figs. 12 and 14 say the opposite, that the plain DLC is very good for wear and smoothness.
Thank you, it was a technical mistake, the figures are revised.
11. Nowhere do the authors state whether the machining was performed dry or with a fluid. This information must be put into the abstract at the beginning so the potential reader will know if it is pertinent to them or not.
In the experiments, the coated end mills were tested without coolant to provide as well better observation of wear, and that correlates to the standard tool tests. The relevant sentence is added to the text of the manuscript.
12. The author’s citation list is very large. Because of this they must know that hard coatings on machine tools is a very well researched topic and the technology is now not only experimental, but commercially available. DLC coatings on end mills are now so common that the reviewer can buy them from a local supply house. The supply house has DLC coated carbide end mills, diamond coated carbide end mills, TiN coated carbide end mills, multi-layer TiN/TiAlN coated carbide end mills, TiCN coated carbide end mills, and plain carbide end mills. Thus the authors must produce a more insightful treatment, instead of using only one kind of DLC, at one thickness, of one composition, at one voltage, at one feed rate, at one speed, for their research. It is quite proper to perform experiments completely with the P1311 unit, but at different thicknesses, or different voltages, or different bond layers, or with different turning speeds. If the reader can make no comparisons then nothing is learned.
All the equations on the thickness of coatings are known and published in manuals from 1975. It is evident that with increasing thickness, the stresses increase, and then there is also milling, in other words, cyclic loads. Those, the film's total thickness for cutters of this diameter does not exceed 4 µm; well, a maximum of 5 µm. Everything that is more will simply peel off. Besides, with less thickness, the effect will not be much. For single-layer DLC coatings, even Platit himself recommends not more than 2 ... 2.5 µm, precisely. Therefore, it only seems that it can vary in thickness. If we would do the job where compared 2 and 4 µm that would be no research since there was everything discovered more than 50 years ago and can be clearly seen from the provided equations. Moreover, with 4 µm, there will be nothing good for sure.
Regarding the fact that the presented studies can have no interest and you can buy a DLC-coated instrument in the store, in the store today, you can buy a wide range of products, including end mills with various coatings. Another issue is that well-known scientific approaches are annually improved. On their basis, new technological solutions are developed. The parameters of the processes of film synthesis are optimized, which differ significantly depending on the coated product's operating conditions. For example, it is the leading manufacturers of cutting tools as Sandvik, Iscar, and others, who annually spend large financial resources on improving coatings. For example, the (TiAl) N coating for cemented carbide was proposed more than 20 years ago, and, nevertheless, even today, its deposition technology is being improved, and it even has the potential for further development.
As for carbon films, today, all over the world, in the framework of the direction of DLC coatings for machine-building applications, many authoritative research teams worldwide continue to work, and interest in them does not decrease, but on the contrary, grows. An illustration of the increasing interest in carbon coatings for mechanical engineering and metalworking needs is the number of scientific publications on this topic (according to the Web of Science database, see the illustration below). If we analyze the Scopus database, there are 2.5 times more such publications over the years. Each research team, depending on their profile and the specifics of the scientific school they represent, carry out research, and publish their results. This article's authors are no exception; we presented original, previously unpublished results focused on practical application in metalworking for end mills that process materials with different properties - aluminum alloys and structural steels. To obtain the results, we used the most advanced equipment, the instrument base of the world's best manufacturers; we obtained good reproducibility of results and conducted a number of tests, including resistance tests (we did not limit ourselves to studying the physical and mechanical properties). We value our reviewer's opinion very much; it is obvious that we are lucky that such a deep and strong scientist was imbued with our work, which undoubtedly has its shortcomings. Something is caused by the fact that we cannot include all the results in the volume of one article; something is connected with annoying errors in the experiment results' graphic design (Figures are provided in the file).
13. Figure 9 is not surprising. Figure 12 is extremely surprising. It is surprising because the better bonded coating performs worse, and also it shows that the cutting performance “sees” what is underneath the DLC coating for steel, but not for Al. This is interesting. Can the authors remark upon this? Is the coating still present or has it been accidently removed?
Thank you that you have noticed it. It was a technical mistake. The figures are revised.
14. The authors need to specify the deposition pressure because this is a fundamental parameter and has an effect on hardness.
Indeed, we did not indicate the pressure at which the formed DLC coatings. We would like to revise this technical mistake, and we agree that this is an important parameter. It should be emphasized that pressure affects the coating properties and is also selected individually for each setup, taking into account its design, particularly the vacuum sensors used and their settings. The manufacturer of our equipment, PLATIT, recommends deposition at a pressure of 4 Pa. It is this pressure that we established during the experiments. We understand why this particular value is recommended. At a pressure below 4 Pa, the deposition rate of carbon condensate drops sharply, which is extremely undesirable for metalworking needs when we need coatings of at least 2 µm, which is quite large thicknesses compared with the needs of microelectronics. At a 4 Pa pressure, a 2.5 µm DLC coating was formed in three hours, which is an acceptable value. With an increase in pressure above 4 Pa, the formed carbon condensate structure changes significantly - the proportion of the graphite-like component begins to prevail in it. For example, as we established in an experiment carried out for curiosity in a laboratory setup, at a pressure of 8 Pa, the coating is almost 90% graphite. Recall that in our case, at a pressure of 4 Pa, we have obtained DLC coating, which consisted of ~ 50% of the diamond component. We believed that these positions are known to specialists, and there is no need to write about them separately in the article.
In the penultimate paragraph of section 2.2. Technology and equipment for the deposition of DLC coatings on end mills we have added information:
The DLC layer was deposited at a pressure of 4.0 Pa. This value's choice is because a decrease in this parameter leads to a decrease in the productivity of the process when an increase leads to an excessive increase in the structure of the graphite-like component's coating.
15. The films themselves are more properly called a-H:C. or perhaps a-C:Si:H, because they are amorphous and they contain much hydrogen and some Si.
In section 2.2, in the second paragraph, the following sentence was added: “Following the generally accepted classification and terminology, the DLC coatings deposited in this work refer to hydrogenated diamond-like a-C: H films.”
16. Fig. 10 is of poor quality. Fig. 13 is a little better.
The figures are revised. We should note that another reviewer prefers yellow color to red. We have changed a part of the text in the image in yellow and added scale bars in white.
17. Most of the introduction must be eliminated. For example, lines 27-42 can be eliminated. Everyone knows that machining and end mills are important. This reads like a popular trade journal, not a research article.
Thank you for your kind comment. The lines are eliminated.
18. Fig. 1 and most of the discussion must be eliminated. The authors are nowhere doing research on DLC, and thus this has nothing to do with the manuscript. The authors are using a single type of DLC and studying flank wear. The workers must concentrate on their findings, and not attempt to teach the basics of DLC structure, which is irrelevant to the paper. On the other hand, if the workers had performed a verity of experiments with different feed gases and targets then this would have been a DLC paper, but they did not do that.
Figure 1 is removed.
19. The authors must detail how flank wear is measured. There seems to be only a few lines of explanation. This is important. Is it the width? The volume? How large an area? Is it proportional to any physical quantity? Is it uniform? Do the authors measure at a specific distance from the cutting edge?
In section 2.1 "Processed materials, cutting tools, and operational testing methods" of the first edition of the manuscript, in the penultimate and last paragraphs, we have already given a description of the method for measuring wear (figure are given in the file.):
«The limit size of the wear chamfer on the flank surface – 0,4 mm-was taken as the criterion for loss of performance (failure) of the end mills. Tool wear resistance was defined as the cutting time until the cutter reached the limit of wear. To quantify wear, we used a metallographic optical microscope Stereo Discovery V12; on its table, the milling cutters were placed in a particular device at an angle of 45°. Each tooth of the cutter was measured, and the most enormous amount of wear was detected, and this was taken into account when processing the results of experiments and constructing curves for the dependence of wear on the flank surface of the cutting time, based on which the resistance value was calculated.
When processing 41Cr4, one pass of the cutter was 3,5 minutes, and after each pass, the wear chamfer was measured. When processing AlCuMg2, one pass of the cutter was 3 minutes, and taking into account that the flank surface wear is less intense when processing aluminium alloys, wear was measured every three passes.»
Besides, in the manuscript, as an illustrative example, we cited two images obtained using an optical microscope (for each type of cutter, there were many such images as they wear out). Based on a variety of measurements, graphs of the dependence of wear on the operating time of end mills were built, which are also given in the text of the manuscript.
The technique is too standard for us to describe in more detail. Nevertheless, if the reviewer has any questions, we are obliged to provide him with explanations that are more detailed.
After each pass, the cutters were sent for measurements to determine dimensional wear along with the flank chamfer. To determine the end mills' wear, we used a SteREO Discovery V12 research microscope located in the materials cutting laboratory of the MSTU Stankin. The microscope has a device for focusing the image visible through the eyepiece. The search for an object in the eyepiece field was carried out by the possibility of displacement in different planes of the object stage.
The cutters were installed in a particular device at an angle of 45° relative to the surface of the measuring device's stage. Measurements were carried out on each tooth of the cutter to identify the most significant amount of wear, which was subsequently taken into account in the work results when constructing the graphs given in the work (an example is given below). Each point on the graph is a single measurement of dimensional tooth wear, as seen through the microscope eyepiece and measured with a ruler.
20. An image should never be referred to as having a certain magnification, for example line 299, because the final printed size is not known. Instead, every image must have a scale bar on it.
This comment is quite strange since it is normal practice to show magnification of microscopic images. We mention it always, and we see other authors do the same. It is quite evident that there are two concepts: objective magnification of the image generate automatically by zooming the object and resolution of the obtained image. Resolution influences the quality of printing (300 dpi in our case). At the same time, you can regulate the size of images in Office Word as you wish with any resolution, and it is not related to the microscopic magnification. A detailed explanation is given in the file.
An example is here, Figure 11 of the manuscript https://www.mdpi.com/2227-7080/8/3/49/htm
We would like to note that another reviewer has asked for added parameters of the microscoping to the figures, including magnification.
21. The reviewer commends the authors for the detailed XPS and flank wear data obtained at frequent intervals. The shape of some of the curves loo similar, almost parabolic. Can the shape be related to a fundamental wear mechanism? Can the shape say anything about carbide vs. coated?
The shape corresponds to the classical curves of the development of wear over time, and the wear mechanism in this speed range is known and it has an adhesive character.
22. 50% sp3 is large. A reader might wonder what kind of DLC is being made. What does the literature say?
You are absolutely right, we should have clarified the type of DLC coatings.
In section 2.2, in the second paragraph, we added as follows:
Following the generally accepted classification and terminology, the DLC coatings deposited in this work refer to hydrogenated diamond-like a-C: H films.
Regarding the question on 50% of sp3 diamond content, it is traditional for such coatings. Following the works of reputable scientists [1-3], when choosing rational deposition modes for such coatings, the formation of a diamond phase is in the range of 30 ... 50%. Thus, our results are at the upper end of the range. We would like to confirm that we research DLC-coatings and produce them for a long time, and therefore our installation works now at its maximum capacity.
In the penultimate paragraph of subsection 3.1., the last sentence we added:
According to authoritative scientists' works, when choosing rational modes of deposition of the investigated type of DLC coatings, the formation of a diamond phase (sp3 hybridization) in the range of 30 - 50% is characteristic [10, 20, 24, 26]. That is, the results obtained are at the upper limit of the possible range of values.
References:
- Robertson J. // Mater. Sci. Eng. 2002 Vol. 37 Р. 129—281.
- Xie Y., Llewellyn R.J., Stiles D. // Wear. 2001 Vol. 250, Р. 88—99.
- Pang X., Yang H., Gao K. et al. // Thin Solid Films. 2011, Vol. 519 Р. 5353—5357.
Reviewer 3 Report
The paper can be published after the following minor revisions:
1. Experimental details (energy of beam, working distance,…) for SEM analyses should be presented.
2. The scale length in SEM images (Figure 5) is not clear. Please check and revise.
3. The quality of Figures 9,11,12,14 should be improved.
4. Conclusions section should be shortened. The authors should better highlight the novelty of this article respect to literature.
Author Response
Response to Reviewer 3 Comments
Dear reviewer,
Thank you very much for your kind evaluation of our work. We do agree with all your proposals and comments and have modified the manuscript according to them.
Introduced changes were marked by blue in the text of the manuscript.
Kind regards,
Authors.
1. Experimental details (energy of beam, working distance,…) for SEM analyses should be presented.
The experimental data is added.
2. The scale length in SEM images (Figure 5) is not clear. Please check and revise.
The figure is revised.
3. The quality of Figures 9,11,12,14 should be improved.
The figure are revised.
4. Conclusions section should be shortened. The authors should better highlight the novelty of this article respect to literature.
The relevant text is revised.